# Deep learning to infer eddy heat fluxes from sea surface height patterns of mesoscale turbulence

Tom M. George[1,2], Georgy E. Manucharyan[1,3 ✉] & Andrew F. Thompson[1]

Mesoscale eddies have strong signatures in sea surface height (SSH) anomalies that are measured globally through satellite altimetry. However, monitoring the transport of heat associated with these eddies and its impact on the global ocean circulation remains difficult as it requires simultaneous observations of upper-ocean velocity fields and interior temperature and density properties. Here we demonstrate that for quasigeostrophic baroclinic turbulence the eddy patterns in SSH snapshots alone contain sufficient information to estimate the eddy heat fluxes. We use simulations of baroclinic turbulence for the supervised learning of a deep Convolutional Neural Network (CNN) to predict up to 64% of eddy heat flux variance. CNNs also significantly outperform other conventional data-driven techniques. Our results suggest that deep CNNs could provide an effective pathway towards an operational monitoring of eddy heat fluxes using satellite altimetry and other remote sensing products.

[1] Division of Geological and Planetary Sciences, California Institute of Technology, Pasadena, 1200 E California Blvd, CA 91125, USA. [2] The Cavendish Laboratory of Physics, University of Cambridge, JJ Thompson Avenue, Cambridge, CB1 3FZ, UK. [3] School of Oceanography, University of Washington, Seattle, WA 98195, USA. ✉email: gmanuch@uw.edu

Energetic, vertically sheared ocean flows, e.g. the Gulf Stream, Kuroshio Current, and Antarctic Circumpolar Current, are baroclinically unstable[1–3], generating mesoscale eddies—vortices with scales of motion of the order of 10–100 km in the open ocean[4,5]. Across most of the ocean, mesoscale eddies make the largest contribution to the kinetic energy and cumulatively dominate the transport of tracers across frontal currents[6–8], though other types of currents such as jets and filaments can also contribute to tracer transport. Surface expressions of mesoscale eddies are monitored globally by satellite altimeters measuring the dynamic anomaly of sea surface height (SSH)[9], which is proportional to the streamfunction of the surface geostrophic flow. Nearly three decades of satellite SSH observations have significantly advanced our understanding of eddy propagation[10], ocean energetics[11–13], and tracer diffusivities[14,15].

Eddy heat fluxes or, more generally, eddy buoyancy fluxes play a fundamental role in eddy-mean flow interactions[16]. The eddies significantly influence the background mean flow by converting available potential energy from the mean currents into eddy kinetic energy (EKE) energy, with the transfer rate being proportional to the average eddy heat flux[17,18]. When the eddy field is spatially heterogeneous, it is only the divergent component of the eddy heat flux that reflects energy exchanges with the mean flow while the rotational component balances the advection of the eddy potential energy (EPE) by the mean flow[19]. The divergent component of the eddy heat flux is also crucial when considering the evolution of the local heat content in the ocean[7]. Despite their profound role in ocean circulation, oceanic mesoscale eddies are not fully resolved in long-term climate projection models due to current computational limitations[20,21] so their impact on larger-scale circulations and tracer fields must be represented in other ways. Expressing the eddy tracer fluxes in terms of mean flow properties is a complex theoretical problem that is yet to be solved, although practically relevant eddy parameterizations have been built and are actively being used in ocean models[22].

Global monitoring of eddy heat fluxes to test and inform physically based parameterizations remains a major challenge for several reasons. Firstly, the eddy flux is difficult to estimate from sparse observations as it is not a sign-definite quantity and its regional or temporal average can be an order of magnitude smaller than its local maxima. Secondly, and more fundamentally, the dynamics of baroclinic instability depend on interactions between the upper and lower layers of the ocean[1] and hence direct calculations of eddy heat fluxes require simultaneous observations of the near-surface horizontal velocity field and the temperature and velocity fields at eddy scales. While surface ocean currents can be globally estimated with satellite altimetry, the ocean interior currents and buoyancy distribution must be observed with in situ instruments. Regionally, some of the most ambitious observational projects in recent years have focused on determining the ocean heat (and volume) fluxes across a fixed latitude or a cross-current transect, including the RAPID[23], OSNAP[24], and cDrake[25] programs. Global in situ observations of subsurface properties, e.g. by ARGO floats[26], moorings[6], and ship transects, remain too spatially and/or temporally sparse to resolve the three-dimensional structure of mesoscale eddies, and thus a direct evaluation of eddy heat fluxes globally is not possible.

Nonetheless, several techniques have been proposed to estimate the eddy heat fluxes using satellite SSH and surface temperature observations along with in situ ARGO float observations by fitting the data to idealized models[27–30]. Other methods also include data assimilation in primitive equation models[31,32] and parameter estimation via stochastic Kalman-type filters in quasigeostrophic models[33]. Because of the relative sparseness of in situ observations, the heat flux estimation techniques must rely heavily on the global eddy-resolving satellite observations of the

ocean surface to approximately constrain the unobserved (or poorly observed) subsurface velocity and buoyancy distributions.

Critically, subsurface flows in the ocean are highly correlated with surface flows, such that the vertical distribution of currents can be represented with a single empirical orthogonal function (EOF) capturing over 80% of the variability[34]. However, the linearly correlated components of the surface and subsurface flows contribute no meaningful domain-averaged eddy heat flux (see "Methods"). An estimation of the eddy heat fluxes requires an accurate measure of the component of the subsurface flow that is spatially uncorrelated from the surface flow. Studies reporting highly skilled reconstructions of mean subsurface flows from mean surface flows[35] or reconstructions of the subsurface flow from SSH and surface temperature observations[36] may, in fact, only be reflections of their high degree of linear correlation, which would not ensure the accuracy of a subsequent estimation of the eddy heat fluxes.

Given the highly nonlinear and chaotic nature of mesoscale turbulence, no theoretical prediction of the surface–subsurface relationship for baroclinically unstable flows currently exists and hence a method for reconstructing heat fluxes from SSH snapshots alone has not yet been developed. Nonetheless, considering the quasigeostrophic (QG) model of eddy formation[18,37], eddy heat fluxes emerge from baroclinic instabilities during which the unobserved bottom flow is affecting, and is affected by, the observed surface flow. Thus, observed eddy patterns in an SSH snapshot could significantly constrain the posterior distribution of subsurface flow and thus contain at least partial information about the corresponding eddy fluxes. This prompts a natural question: how much information is contained in the SSH field with regard to the eddy heat flux estimation?

Here we address this question by considering data-driven approaches based on deep Artificial Neural Networks (ANNs)[38], which are powerful tools for extracting critical, if subtle, information from large volumes of data[39–44]. ANNs are widely used for supervised learning tasks where an approximation of an input-to-output mapping can be iteratively developed by optimizing a highly nonlinear function with respect to a large number of trainable parameters. Specifically, in fluid mechanics, deep neural networks have been used to address the closure problem in Reynolds-averaged Navier–Stokes equations[45–47] outperforming other data-driven methods such as dimensionality-reduction via proper orthogonal decomposition[48] or dynamic mode decomposition[49,50]. For geostrophic turbulence[37], ANNs and, more specifically, convolutional neural networks (CNNs) have been used to demonstrate a strong potential for parameterizations of eddy momentum fluxes in barotropic[46,51] and baroclinic[35] ocean gyres. In theory, deep ANNs can approximate nonlinear mappings of any complexity (see the universal approximation theorem[52,53]), provided the network contains a sufficient number of free parameters, and there exists a sufficient amount of training data.

In this study, we assess the plausibility of predicting the instantaneous domain-averaged eddy heat flux by extracting information only from the SSH patterns of mesoscale eddy field. Since the variability of large-scale oceanic flows is predominantly contained in the barotropic and the first baroclinic modes[54–57], our research philosophy here is to use one of the most fundamental and influential models of baroclinic turbulence—the two-layer QG model[2]. Despite being idealized, the QG model exhibits nonlinear chaotic behavior and can have symmetry breaking multiple equilibria[58,59]. The QG model allows us to estimate the heat flux predictability limit free from other practical constraints such as the number of available samples, their spatial sparseness, measurement inaccuracy, and external noise. We quantify the limits of the predictive capabilities of SSH data for diagnosing

eddy heat fluxes by using data-driven approaches (CNNs among others) trained on large volumes of data from the eddy-resolving QG simulations. We will demonstrate that CNNs are powerful tools for extracting the desired information from SSH snapshots by identifying the spatial patterns containing the most relevant information for flux predictions. Finally, we will put forward a hypothesis that there exists an upper bound in the predictability of eddy heat fluxes based only on SSH snapshots of baroclinic ocean turbulence.

## Results

**Eddy heat fluxes in geostrophic turbulence**. We conduct idealized numerical simulations of a two-layer quasigeostrophic model of mesoscale turbulence with a prescribed baroclinically unstable background flow that is horizontally uniform, vertically sheared, and kept constant in time (see Methods for model equations and parameters). The simulations are performed using spectral methods within a large, 4000 km × 4000 km, doubly periodic domain containing about 100 Rossby deformation radii per side. After the initial spinup phase, the baroclinic turbulence equilibrates and mesoscale eddies become prominent throughout the domain (Fig. 1a). Cyclonic and anticyclonic eddies are clearly visible in the SSH snapshots and have pronounced filaments of potential vorticity at their edges due to the enhanced spatial structure in the vorticity field (Fig. 1a, b). The dynamical variables (potential vorticities and streamfunctions) contain signatures of eddies and filaments that appear to be correlated between the two layers (Fig. 1b); yet, it is the decorrelated components of the two streamfunctions that are associated with the eddy heat flux (Methods). The averaged eddy heat fluxes are directed in such a way as to induce an overturning circulation that tends to flatten the tilted thermocline and reduce the magnitude of the mean shear (Fig. 1c), although in this model the vertically sheared mean flow is fixed and acts as a perpetual source of energy.

In regions with significant spatial heterogeneity in the EPE distribution, it is only the divergent component of the eddy heat flux (as opposed to the rotational component) that is associated with eddy-mean flow energy exchanges[7,19]. The divergent and rotational components of eddy fluxes can be estimated using the Helmholtz decomposition, with an important caveat that the decomposition is not unique as it requires specification of a priori unknown boundary conditions, and even for doubly periodic domains used in our study, the decomposed fluxes are defined up to an arbitrary constant. Because our simulations are performed in a doubly periodic domain with homogeneous background flow, the distributions of eddy kinetic and potential energy are statistically homogeneous throughout the domain and the eddy heat flux contains a significant uniform component in the down-gradient direction. In the absence of a statistically significant advection of EPE by the mean flow and by the eddies, the rotational flux in our simulations is negligible, at least when the subdomain over which the average is taken is sufficiently large. Thus, the spatially uniform component of the eddy heat flux in our simulations should be interpreted as the divergent flux. As such, we define subdomains (1000 km × 1000 km) within a doubly periodic simulation domain (Fig. 1a) to build a dataset of nearly independent eddy field realizations and we estimate the subdomain-averaged eddy heat fluxes as they directly affect the evolution of the subdomain EKE (Fig. 1). The subdomain fields are no longer doubly periodic and hence individual eddies passing through the boundaries can lead to non-zero instantaneous heat flux divergences that we also strive to predict with deep learning.

The eddy heat fluxes fluctuate dramatically on monthly timescales, ranging in magnitude from nearly zero to over double their mean values (Fig. 1d). The decorrelation timescale for the

heat flux time series,~20 days, is roughly half the period between SSH snapshots (Fig. 1e), implying that a subtle change in SSH patterns result in a significant change in the eddy heat flux. The high sensitivity to SSH patterns reflects the fact that eddy heat fluxes are proportional to a correlation between surface velocity, which is directly related to SSH (see Eqs. (3) and (4)), and an unknown subsurface streamfunction (Eq. (5))—both of which evolve differently according to a set of strongly nonlinear but coupled equations (Eqs. (1) and (2)). It is the lack of any explicit information about either subsurface flow or (equivalently) thermocline depth anomalies that makes the problem of eddy heat flux mapping from SSH data mathematically ill-defined.

The subdomain-mean EKE lags the eddy heat flux by about 10 days (Fig. 1e), supporting the notion that the presence of the down-gradient eddy heat flux during baroclinic instability enhances EKE through eddy generation. Since the background mean flow in our model does not change during the EKE and heat flux fluctuations, it is not possible to express the eddy fluxes in terms of the mean flow over timescales of months and shorter. Yet, ocean EKE fluctuations on these timescales are critical for interior diabatic mixing as well as coupling with surface forcing from the atmosphere and cryosphere. At these relatively short timescales, mean flow observations alone are insufficient and we must search for ways to estimate the time varying components of the eddy heat flux from the information we have available at eddy scales. We thus aim to identify the existence of a statistical relation between the domain-averaged eddy fluxes (panel d) and the corresponding SSH snapshots (Fig. 2b, top right). Note that estimating instantaneous eddy fluxes presents a significantly less-constrained problem compared to the conventional eddy parameterization problem of reconstructing long-term mean eddy fluxes, which are indeed expected to depend only on the mean flow.

**Deep CNNs predict eddy heat fluxes from SSH snapshots**. Here we discuss the CNN architecture used in this study and its skill at predicting instantaneous domain-averaged eddy heat fluxes given only SSH snapshots. Historically, CNNs have been successful at solving image/pattern recognition problems for which no analytical solution exists[60]. Like all "deep" neural networks, CNNs are built from many simple layers stacked atop one another, not necessarily in a sequential order. Each convolutional layer filters the output of the layer before by convolving a small filter matrix across it, applying a predefined nonlinear activation function, and performing batch normalization. Both the depth and non-linearity of the resulting model are key to explaining its strength. CNN filters are not specified a priori but instead are optimized using the input/output data until they minimize an objective error function using some version of a gradient descent algorithm[61].

The moderate complexity CNN architecture, mapping the input (SSH snapshot) to the output (eddy heat flux), is conceptually shown in Fig. 2a. It consists of three convolutional layers followed by two fully connected layers. Each convolutional layer filters the output from the preceding layer (Fig. 2b, left two columns) by convolving them with small 4×4 weight matrices (Fig. 2b, rightmost column). The information extracted in the last convolutional layer is then passed to a fully connected neural network consisting of two hidden layers that map to the output. The output from each set convolutional filter is passed through a nonlinear activation function[62] (ReLU) before being fed to the next layer. In total, the CNN has $O(10^5)$ free parameters that are iteratively updated using a stochastic gradient descent method[61] to maximize the flux prediction skill. It takes $O(10)$ iterations through the training data (aka epochs) of $O(10^5)$ images to find

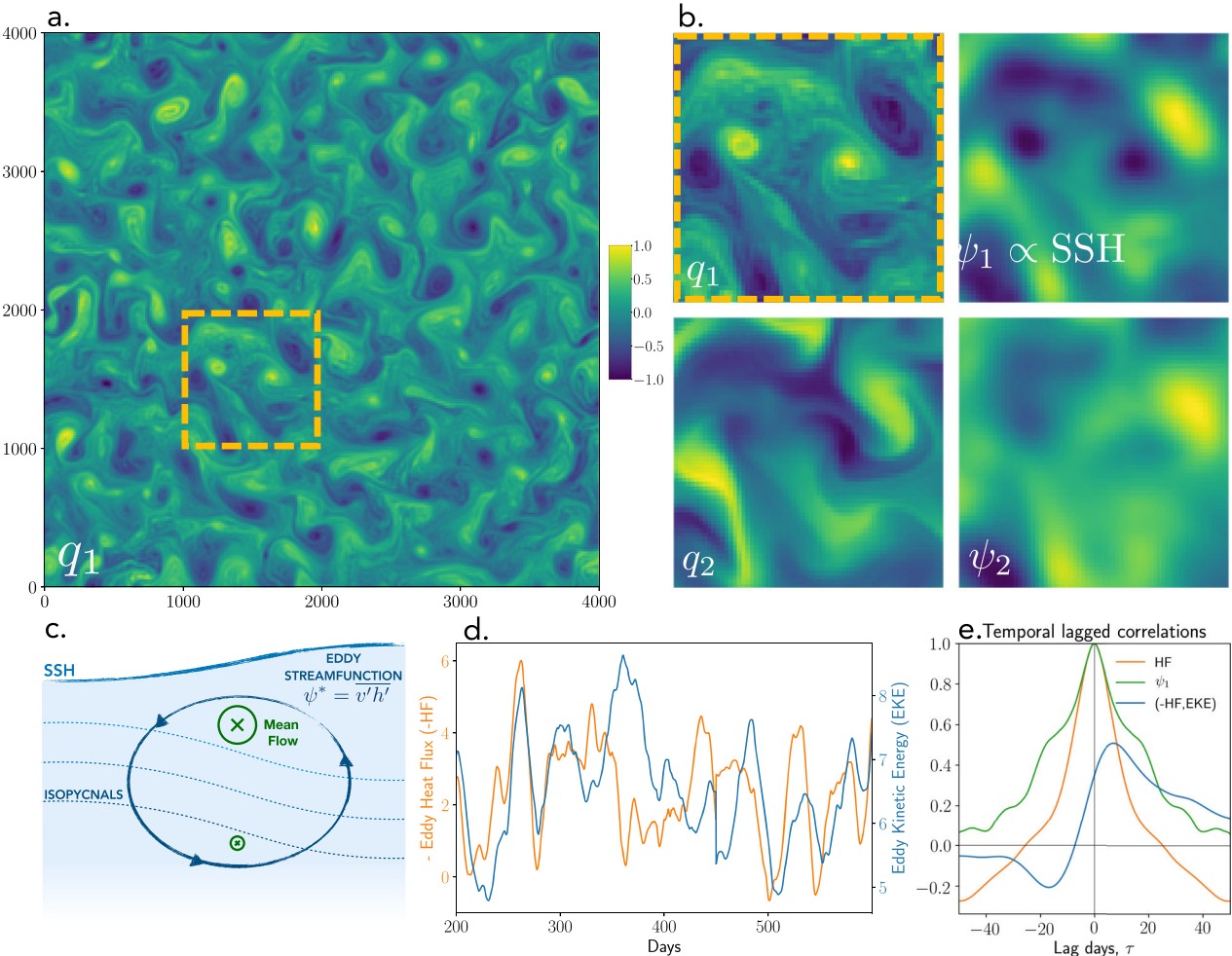

**Fig. 1 Simulations of baroclinic geostrophic turbulence on a β-plane. a** Snapshot of the upper-layer potential vorticity field, $q_1$, showing the prominence of mesoscale eddies; orange dashed square shows a representative $1000 \times 1000$ km subdomain. **b** Subdomain snapshots of potential vorticities $q_i$ and streamfunctions $\psi_i$ in the upper ($i = 1$) and lower ($i = 2$) model layers. Note, the sea surface height (SSH) is proportional to the top layer streamfunction, $\psi_1$. All variables have been non-dimensionalised such that their extremal pixel has a value of modulus 1. **c** Cross-sectional schematic of a stratified ocean, demonstrating the thermocline tilt and corresponding vertically sheared mean flow (for the northern hemisphere); the cumulative impact of all eddies creates an overturning eddy streamfunction, $\psi^\star$, that acts in a direction to flatten the thermocline slopes and decelerate the mean flow. **d** Temporal evolution of the subdomain-averaged eddy heat flux $HF$ (it's negative is plotted with the orange curve) and the domain-averaged eddy kinetic energy $EKE$ (blue) demonstrating strong fluctuations within the statistically equilibrated mesoscale turbulence on monthly timescale. **e** Autocorrelation functions for the eddy heat flux (orange), sea surface height snapshots (green), as well as the lagged correlation between the negative eddy heat flux and the eddy kinetic energy (blue), with positive lags corresponding to the eddy heat flux preceding the eddy kinetic energy. Note that the eddy fluxes decorrelate almost twice as fast compared to the sea surface height snapshots and the heat flux is an early predictor of the eddy kinetic energy.

the optimal set of parameters (Fig. 2c). It is essential to ensure that training and testing data samples are completely independent of each other to avoid overfitting, which we do by using two independently noise-seeded numerical simulations.

By optimizing information extraction from SSH snapshots, the learned filters reflect dynamically relevant features. For example, the set of filters in the first convolutional layer, $f_i$, can be split into two representative groups identifying cyclones, e.g. $f_1, f_2$, and anticyclones, e.g. $f_3, f_4$, (Fig. 2b, right two columns), while slight filter differences ($f_1 - f_2$ and $f_3 - f_4$) emphasize eddy gradients and edges, particularly for dipoles (Fig. 2b, middle column). From linear stability analysis, the eddy heat fluxes should depend on the relative position and strength of eddies in both layers, with the magnitude of the flux being particularly strong in baroclinic dipoles known as hetons[63]. Thus, it is reassuring that the network has learned to extract this type of information from SSH

snapshots. In subsequent layers the information becomes too abstract for interpretation. The average testing skill (defined in Eq. (6)) achieved by the CNN peaks at 0.36 (Fig. 2c), corresponding to a relatively high correlation of 0.8 between the predicted and true eddy fluxes. The CNN is highly efficient at extracting the required information from SSH patterns, so much so that it outperforms other tested data-driven methods that either disregard the two-dimensional nature of SSH data or attempt to use more simplified linear methods (see Methods).

Despite explaining up to 64% of the eddy heat flux variance (Fig. 3a, $R^2 = 64\%$ was the maximum achieved value), CNN predictions have some systematic biases reflecting the fundamental limitations of the information contained in SSH snapshots. Firstly, extreme values of eddy heat fluxes (over one or two standard deviations from the mean) are persistently underestimated by the CNNs (Fig. 3b, d). This underestimation

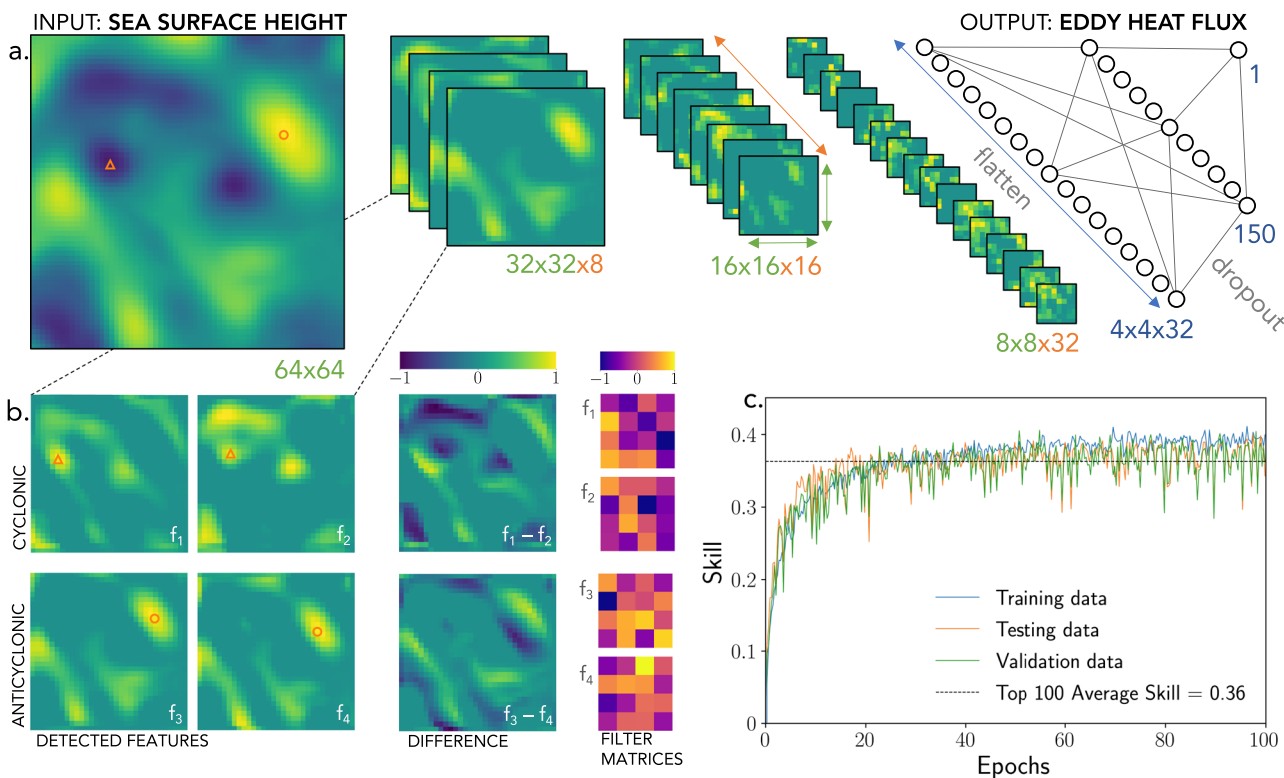

**Fig. 2 The convolutional neural network architecture optimized to predict eddy heat fluxes from sea surface height snapshots. a** The architecture consists of multiple layers that consecutively filter the input, by convolution, with 4×4 weight matrices. The size of the feature maps (green numbers) is decreased layer-by-layer whilst the number of feature maps (orange numbers) is increased. More and more abstract features of sea surface height patterns are extracted at each subsequent layer. Finally, the features extracted by the final layer of convolutional neural network are flattened into a vector and passed into a two-layer fully connected neural network in order to predict the flux. All variables have been non-dimensionalised such that their extremal pixel has a value of modulus 1. **b** Example of the outputs from the first convolutional layer after training. The left two columns, $f_i$, show 4 of the 8 first layer feature maps—we discover that they are split into cyclonic and anticyclonic filter groups. Also shown are differences between these feature maps (third column) as well as the corresponding 4×4 weight matrices (fourth column) which performs this filtering. The triangle/circle markers highlight example cyclonic/anticyclonic eddy centers. **c** The evolution of the neural network skill evaluated using the training (blue), testing (orange), and validation (green) datasets, plotted as a function of training epochs, i.e. the number of times we have iterated through the entire training dataset; the average validation skill corresponding to the top 100 training skills is shown with dashed gray line.

remained true even when (i) synthetically increasing the number of training examples with extreme eddy fluxes and (ii) testing various optimizers (Stochastic Gradient Descent, Adam, Adamax[64]), losses (mean absolute and square error) and weight regularizations (L1 and L2). This suggests that deficiencies of the CNN architecture or lack of training data do not explain the limited skill, but rather this is caused by the inherent incompleteness of the information contained in SSH snapshots. Secondly, since the CNN was trained on nearly decorrelated SSH snapshots separated by 10 days, evaluating its performance on continuous SSH time series generates elevated variability at timescales shorter than about 10 days (Fig. 3c). Still, the CNN generates a skillful and relatively smooth eddy heat flux time series (Fig. 3d) with similar statistics to the true flux. While superior network architectures that would eliminate these biases might exist, it is not evident they could achieve a significantly higher predictive skill. Indeed, significantly increasing the number of training samples or the network complexity does not indicate improvements in the skill (Fig. 3e, f), implying an underlying limitation of the chosen architectures or a theoretical (dynamically constrained) upper bound associated with the information contained in SSH snapshots. Nevertheless, as Fig. 3d demonstrates, even with the skill of about 0.36, CNNs can provide valuable information on the eddy heat flux variability on monthly timescales.

**Prediction of eddy heat flux divergence.** For complex oceanic flows, the divergent component of the heat flux, which governs the energy exchange with the mean flow, can be much smaller than the total flux due to a large rotational component[7]. The divergent component of the flux is commonly calculated from the eddy heat flux using the Helmholtz decomposition that requires computing the heat flux divergence (HFD). The HFD is also a desirable quantity to estimate as it directly affects the evolution of the local heat content. While the HFD, and hence the divergent component of the flux, can be calculated after estimating the flux using deep learning, the associated errors can be large. Thus, an additional stringent test for the CNNs is to predict the HFD (instead of the heat flux) directly from the SSH data. Here we show that CNNs can indeed be trained to learn the heat flux divergence outright.

We calculate the HFD as an average within an inset region with boundaries separated from the outer boundaries of the SSH snapshot by a distance $x$ (Fig. 4a). Similar to the heat flux, the HFD depends on the lower layer streamfunction and so cannot be calculated analytically. The same neural network as displayed in Fig. 2 is trained to estimate the HFD given the corresponding SSH snapshot. With a boundary inset, $x = 200$ km, the CNN can learn to estimate the HFD with an average skill of 0.35 (Fig. 4b). The number of required epochs as well as the resulting skill are similar to the task of predicting the heat flux (compare Figs. 4b and 2c).

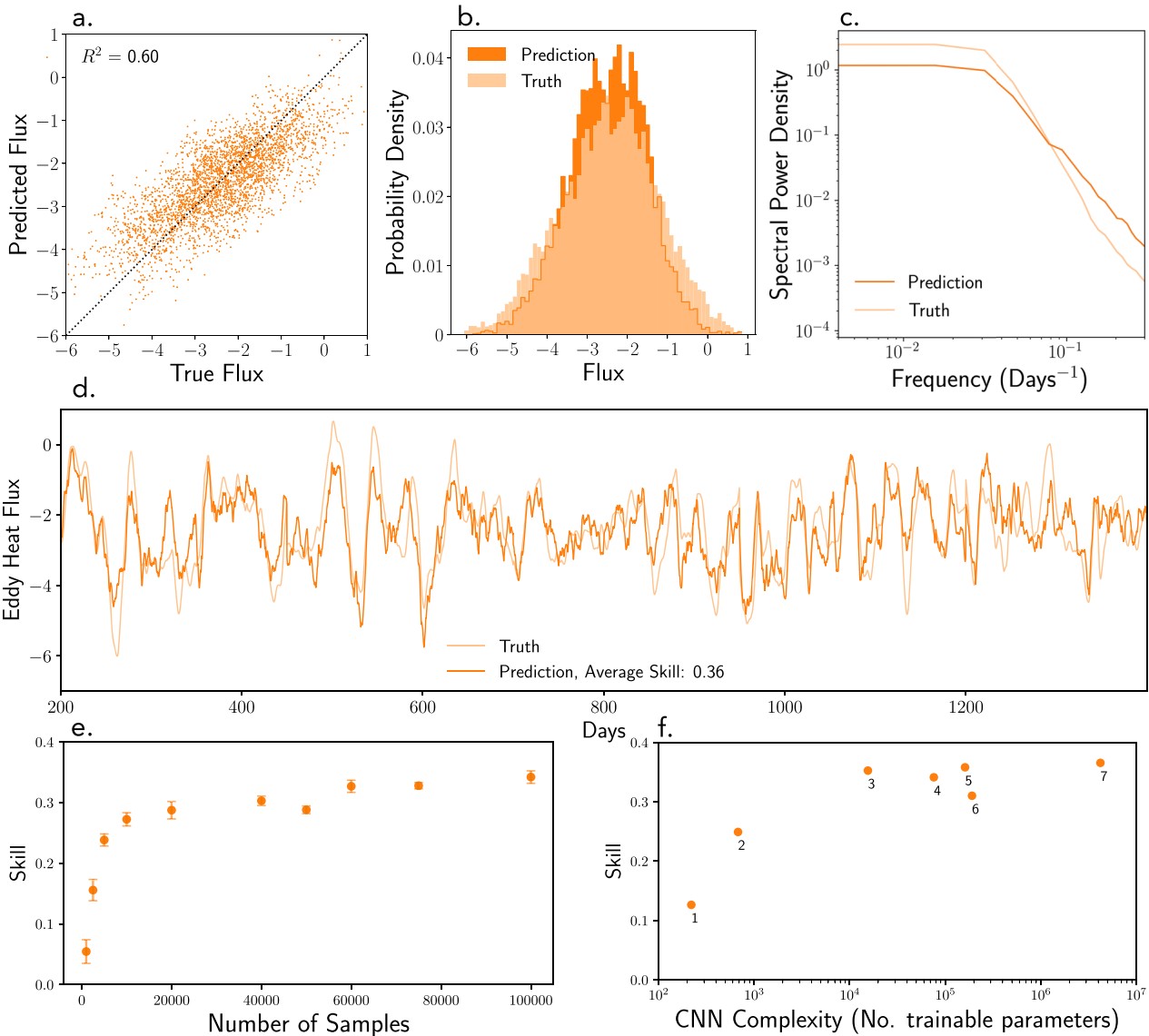

**Fig. 3 Performance of convolutional neural networks in estimating instantaneous domain-averaged eddy heat fluxes from SSH snapshots.** Panels **a–d** show comparisons between the predictions (dark orange) and the true values (light orange) for various performance diagnostics. **a** Scatter plot of predictions for the validation dataset demonstrating that convolutional neural networks explain over 60% of the flux variance (max achieved $R^2 = 0.64$). **b** Histograms highlighting the prediction biases towards underestimating extreme values of eddy fluxes. **c** Power spectra highlighting the prediction biases towards producing noisier time series at frequencies higher than $O(0.1 \text{ days}^{-1})$. **d** Time series showing the fluctuations of the true and predicted eddy heat fluxes, highlighting the skill and relative temporal smoothness of the prediction as well as its deficiencies on individual events. The eddy heat flux prediction is plotted for the network state with maximum validation skill (skill metric defined in Methods), corresponding to a test data skill of 0.36. **e** Sensitivity of the fully trained convolutional neural network skill on the number of data samples used for training; error bars represent a range in the validation skill corresponding to the top 100 best training skills. **f** Prediction skills achieved for a variety of convolutional neural network (CNN) architectures ranging from simplistic to deep; the number of adjustable parameters is shown on the x-axis as a rough measure of CNN complexity while hyperparameters are referred to by numbers and described in Methods.

We do not observe divergent overfitting over the full training period of 100 epochs. For small values of $x$, or when the HFD boundary is close to the edge of the SSH subdomain, the CNN struggles to learn the HFD (Fig. 4c). The skill improves once the boundary inset $x$ is increased, achieving an optimum when $x \approx 200$ km from the edge of the subdomain (Fig. 4c). The interpretation is that eddies crossing the boundary of the divergence region are responsible for the HFD fluctuations and hence their SSH expressions must be "visible" to the CNN as input, i.e. the boundary inset $x$ should be larger than a characteristic eddy size. The CNN skill does not deteriorate as $x$ increases towards 500 km and the inset region shrinks to a point

(Fig. 4c), implying that point-wise estimation of heat flux divergence is possible with the same accuracy as for averages over large domains. Evaluating the performance on an example of continuous SSH evolution, the trained CNN performs well in predicting the monthly variability of the heat flux divergence (Fig. 4d), with large errors apparent only for a few rare cases.

**Optimal CNN complexity and required volume of data.** To identify and prevent overfitting, regularization techniques, such as splitting the data into independent training/validation/testing sets, applying random dropout to neurons, early stopping of

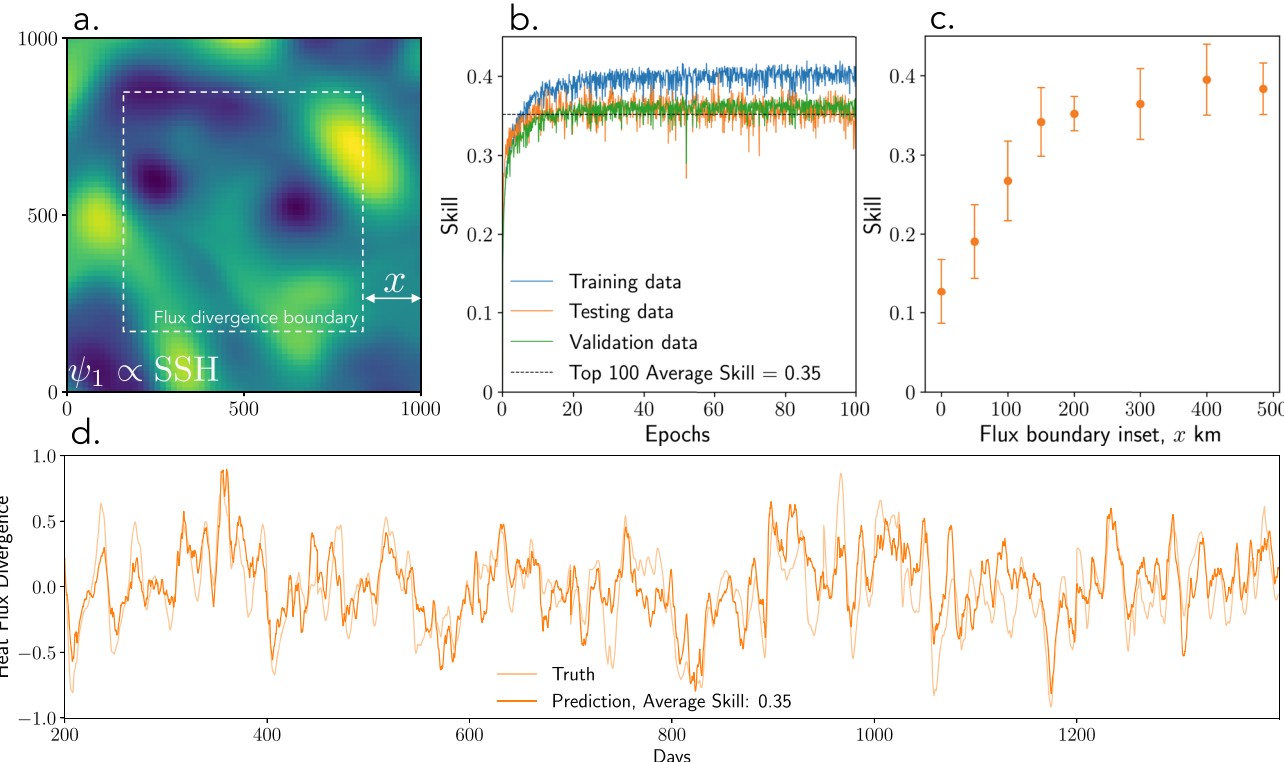

**Fig. 4 Performance of the convolutional neural network in predicting the divergence of the eddy heat flux. a** An example of an input sea surface snapshot and the inset region separated by a distance $x$ over which the output is calculated as the inset-averaged divergence of the eddy heat flux; colorscale and range is the same as in Fig. 1. **b** The evolution of the training (blue), validation (green), and testing (orange) skills as a function of training epochs for the case of $x = 200$ km. The average validation skill corresponding to the top 100 best training skill is shown in gray dashed line. **c** The sensitivity of the prediction skill on the proximity ($x$) of the inset to the boundaries of the sea surface height domain; note that $x \approx 500$ km corresponds to point estimate of the eddy heat flux divergence. The error bars represent a range in the validation skill corresponding to the top 100 best training skills. **d** Example time series showing the fluctuations of the true (thin orange line) and predicted (thick orange line) heat flux divergence for the case of $x = 200$ km.

training and L2 weight regularizations have been applied during CNN optimization. For any neural network, insufficient training data can lead to overfitting and/or skill reduction. Thus the optimal amount of data necessary to achieve the maximum skill depends on the network architecture: higher complexity networks having larger number of trainable parameters can generally achieve higher prediction skills but require larger volumes of training data. Specifically, for the CNN architecture used here, we find that it is necessary to have at least 20,000 SSH images (and their corresponding eddy heat fluxes) in order for training to achieve the maximum skill and to avoid significant overfitting (Fig. 3e). When it comes to using real ocean data to train a neural network, even though the required number of SSH snapshots may be available from satellite altimetry databases, a severe lack of spatially and temporally dense interior ocean measurements means there is little chance of being able to calculate the corresponding heat fluxes. Simply put, the required number of training samples is currently too large to make practical progress and other, more efficient, network architectures must be considered to reduce this number.

We could not construct a CNN architecture that could significantly surpass the skill of 0.36, even when a total of 200,000 training samples were used on ultra-deep CNNs trained intensively on GPUs. Instead, we find that there exists an optimal CNN complexity for this problem: simpler networks cannot achieve the highest possible skill, while complex networks struggle with overfitting and computational cost (Fig. 3f). The optimal CNN architecture (Fig. 2a) still involves a large number

of trainable parameters, $O(10^5)$, and hence is likely to be sufficiently powerful in recovering the physically constrained dependencies between eddy fluxes and SSH snapshots, were they to exist. Here, "optimal" refers to the fact that it performed approximately as well as ultra-deep ResNet CNN (see "Methods") trained for $O(1000$ cpu hours) but required $O(1000)$ times fewer parameters and only $O(1$ cpu hour) of training time. The existence of the upper bound in skill confirms our expectations that there may be process-based limitations on the information contained in SSH snapshots with respect to identifying subsurface flows. Nonetheless, the CNNs explain over 64% of the eddy heat flux variance, performing substantially better as compared to other statistical methods including linear regression, principal component analysis, support vector machines, or random forests (Fig. 5).

## Discussion

Our idealized study provides a proof-of-concept that deep learning could be used for estimating eddy heat fluxes from satellite altimetry. By training deep neural networks on synthetic data from eddy-resolving simulations of baroclinic turbulence, we showed that the eddy expressions in SSH snapshots contain sufficient information to estimate instantaneous domain-averaged eddy heat fluxes and their divergences, accounting for about 64% of their variance. We found that CNNs that explicitly rely on two-dimensional pattern analysis substantially outperform other conventional data-driven techniques, including principle

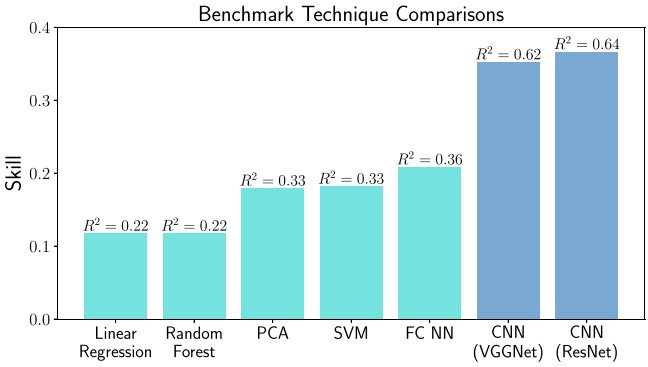

**Fig. 5 Performance chart for a set of various data-driven techniques used to estimate eddy heat fluxes from sea surface height snapshots.** The $y$-axis represents their obtained skill and each column has a denoted value of $R^2$ that reflects the fraction of the total eddy flux variance explained by the fit. The following techniques are shown: Linear Regression, Random Forrest, Principle Component Analysis (PCA), Support Vector Machine (SVM), Fully Connected shallow Neural Network (FC NN), Convolutional Neural Networks with sequential propagation (VGGNet) and with residual connections (ResNet). The techniques are described in Methods. Convolutional neural networks (dark blue) significantly outperform other methods (cyan) as they are optimized to extract the most informative eddy patterns. Note that the FC NN and VGGNet training time is O(1 cpu hour), whereas ResNet takes O(1000 cpu hours) to train.

component analysis, linear regression, random forests, statistical vector machines, and basic fully connected neural networks. This suggests that information about subsurface ocean flows may be linked to the geometric shape of eddies and their relative orientations with respect to each other. For more complex, and thus more realistic, mean flow configurations, e.g. a meandering jet, EPE advection will be significant and eddy heat fluxes would be dominated by the rotational component. In these cases, it will be important to develop neural networks that can distinguish and predict the divergent component of the flux.

Tested over a broad range of network architectures, domain sizes and training dataset volumes, there seems to be an upper bound to the predictive power of the considered CNN architectures—an indication that we may have approached a dynamics-based bound on the information content in SSH snapshots. The existence of this bound would imply that analytical laws directly derived from the QG equations linking eddy heat fluxes to SSH snapshots do not exist, otherwise deep neural networks would have approximated these laws with sufficient amount of training data. In this case, only data-driven regression or probabilistic models could be relevant. It is plausible that the upper bound on the predictive capability of SSH snapshots obtained from the CNNs used in this study may be surpassed by other machine learning architectures, e.g. cross-modal CNNs[65], ultra-deep CNNs[66], or more novel deep learning architectures. Thus, the existence of the ultimate dynamics-based upper bound remains an open question. To encourage the participation of a broader scientific community in addressing this question, we have made the training/testing datasets as well as the CNN code publicly available and we welcome attempts to improve upon the prediction skill found in this study.

Despite the seeming existence of the upper bound, the performance reached by CNNs on the model data is sufficiently high to potentially provide valuable information on eddy-mean flow interactions globally for the ocean, if the deep learning technique could be extended to satellite observations. However, there are several remaining issues that must be addressed before implementing this approach operationally using satellite SSH

observations. Firstly, even within our controlled numerical experiments that are free of external processes, the amount of training data necessary for the supervised learning of CNNs is substantial, $O(10^5)$ samples. Secondly, our idealized simulations of baroclinic turbulence were performed with constant mean flow and stratification parameters, thus ignoring spatially and temporally non-local eddy processes that may complicate the CNN learning and require an even larger volume of training data. Future studies may consider reconstructing eddy heat fluxes in realistic spatially heterogeneous mean current flows by training the neural network on examples from eddy-resolving ocean circulation models. Thirdly, in practice, directly measuring eddy heat fluxes for supervised learning would require an eddy-resolving network of ocean instruments such as ARGO floats, gliders, or moorings that are expensive to sustain on basin scales and over long periods of time. Thus, it is necessary to develop superior deep learning techniques, potentially implementing meta-learning strategies[67–69], which could reduce the volume of necessary training data by at least an order of magnitude. Another way to approach the limitations of obtaining the training data is to explore the possibility of transfer learning[70]: here CNNs could be almost entirely trained on a large volume of synthetic model data (aka. this study) and afterwards use a much smaller fraction of real ocean observations for validation and fine-tuning of weights and biases.

## Methods

**Numerical simulations of geostrophic turbulence.** Baroclinic turbulence was simulated using the two-layer QG equations with a vertically sheared, horizontal uniform background mean flow that is kept constant in time: an idealized view of the baroclinic instability known as the Phillips model[2,18]. The model assumes conservation of potential vorticity in both layers with their anomalies from the mean state, $q_{1,2}$, defined as

$$q_1 = \nabla^2 \psi_1 + \frac{f_0^2}{g'H_1}(\psi_2 - \psi_1), \quad q_2 = \nabla^2 \psi_2 + \frac{f_0^2}{g'H_2}(\psi_1 - \psi_2). \quad (1)$$

The time evolution of potential vorticity anomalies is governed by lateral advection due to eddies, the mean flow and bottom layer Ekman drag:

$$\partial_t q_i + U_i \nabla q_i + \beta v_i = -r_{Ek}\delta_{i2}\nabla^2 \psi_i, \quad i = \{1,2\}. \quad (2)$$

Here $\psi_i$ is the perturbation streamfunction defined by

$$\begin{pmatrix} u_i \\ v_i \end{pmatrix} = \begin{pmatrix} -\partial_y \psi_i \\ \partial_x \psi_i \end{pmatrix} \quad (3)$$

where $u$ and $v$ are the zonal and meridional components of velocity and $r_{Ek}$ represents the Ekman drag coefficient. Simulations are performed for characteristic parameters of a midlatitude baroclinic current such as the Gulf Stream or the Antarctic Circumpolar Current: Coriolis parameter $f = f_0 + \beta y$ (evaluated at 40 degree latitude) and the stratification parameters are chosen to result in a baroclinic Rossby deformation radius of 40 km. The ratio of the top to bottom layer thickness is chosen to be 1:5. The background mean flow is uniform and constant in time, with the vertical shear of $U_1 - U_2 = 0.2$ m s$^{-1}$ being sufficiently large to develop baroclinic instabilities that reinforce generation of strongly interacting mesoscale eddies equilibrating to be O(200 km) in diameter (Fig. 1a). The model dissipation is due to the bottom Ekman drag (10 day timescale), while small-scale vorticity gradients are arrested by a scale-dependent dissipation implemented as a filter in spectral space that damps high wavenumber energy in all model variables each time a Fourier transform is used to evaluate tendencies. These specific parameters were chosen so as to give statistically steady 'weak $\beta$-plane turbulence' corresponding to a midlatitude ocean. There is no reason to believe our machine learning method would not work equally well for other parameter sets, so long as the nature of the turbulence does not change.

The doubly periodic domain was set to 4000 km in horizontal scale and the model equations are solved in spectral space using 256 Fourier modes in both directions. The double-periodicity means the full domain has no overall heat flux divergence. Furthermore, the 4000 km domain was arduous to train using our CNN technique. To overcome these issues the doubly periodic full domain was divided into into 16 subdomains, 1000 × 1000 km each; these subdomain SSH snapshots used for training are no longer doubly periodic and hence the corresponding eddy heat fluxes do have a divergence contribution due to eddies at the boundaries. In total, 112,000 training data images were obtained from 4 independent simulations, each initialized with independent noisy initial conditions and summing up to about 3000 years of model time (spinup data was discarded). Since SSH snapshots decorrelate from themselves over a timescale of 20 days, the time gap between successive training snapshots was set to 10 days to avoid redundant data and to be more in line with real

altimetry data from satellites that have return periods of O(10 days). For training, the 112,000 subdomain snapshots are treated as spatially and nearly temporally independent examples of typical eddy patterns. 16,000 test data images were produced from an independently seeded simulation each separated from the next by $\frac{1}{4}$ day to evaluate the smoothness of the obtained CNN mapping. Note we also tried training the CNN on doubly periodic 1000 km domains and achieved similar results—all other discussions in this paper relate to training on the 1000 km non-periodic subdomains, since performance did not substantially decrease and, in practice, having non-periodic domains would allow us to address the problem in real oceans where domains are not doubly periodic.

The eddy heat fluxes, HF, are defined as $\mathrm{HF} = \overline{v_1 h_1}$, where the overline corresponds to averaging over the subdomain area, $v_1 = \partial_x \psi_1$ is the anomalous surface ocean velocity in the meridional direction perpendicular to the mean flow, and $h_1 = (f_0/g') \cdot (\psi_2 - \psi_1)$ is the thermocline depth perturbation. Note that SSH perturbations are directly related to the surface geostrophic streamfunction as

$$\mathrm{SSH} = \frac{f_0}{g'} \psi_1, \tag{4}$$

and hence the eddy heat flux can be split into the 'trivial' component that only depends on the known SSH field and the 'coupled' component that depends on the unknown bottom layer streamfunction:

$$\mathrm{HF} = \frac{f_0}{g'} \overline{(\psi_2 - \psi_1)\partial_x \psi_1} = \underbrace{\overline{\psi_2 \partial_x \mathrm{SSH}}}_{'\mathrm{Coupled\ flux}'} - \underbrace{\frac{g'}{f_0} \overline{\mathrm{SSH}\partial_x \mathrm{SSH}}}_{'\mathrm{Trivial\ flux}'}. \tag{5}$$

The 'trivial' component of the eddy heat flux exists solely due to eddies passing through the artificially defined subdomain boundaries and it is identically zero in a periodic domain. Not only is this component dependent only on the known SSH field, and hence is trivially calculated, but it is dynamically irrelevant in the sense that it is a noisy term, highly dependent on the location of subdomain boundaries rather than on fundamental processes going on inside it. However, given that the dynamically relevant component of the eddy heat flux (the 'coupled flux') depends on the average of a product between $\psi_1$ and a horizontal derivative of $\psi_2$, it is clear that a component of $\psi_2$ that is proportional to $\psi_1$ also only provides a dynamically irrelevant (and trivial) noisy contribution to the heat flux and does not reflect the intensity of baroclinic instabilities. Thus, a dynamically meaningful heat flux exists only due to a component of $\psi_2$ that is decorrelated from $\psi_1$. The "coupled" flux is also affected by the boundary effects but it nonetheless contains the critical contribution from the fluxes emerging due to baroclinic instability. We thus focus on the prediction of the 'coupled' component of the medidional upper-layer heat flux from SSH snapshots, noting that the 'trivial' component could be exactly calculated from SSH data and added if necessary to give an estimate of the whole flux; we choose not to include the 'trivial' component in the calculation of the prediction skill because this would artificially increase it.

It is well known that the total heat flux can be partitioned into divergent and rotational components (as opposed to our 'coupled' and 'trivial' components), of which only the divergent component contributes to the heat flux divergence[71]. Here we reconstruct the total heat flux—which amounts to predicting the coupled component—from which the heat flux divergence (or any other quantity) can then be determined. Note, it is only the heat flux divergence that contributes to a local heat flux tendency.

Finally, for any given subdomain, SSH snapshots and corresponding eddy heat fluxes are then used as training inputs and outputs for the data-driven mapping methods. Importantly, we aim to predict the instantaneous flux given an instantaneous SSH snapshot and, although the CNN results are evaluated on continuous timeseries (Fig. 3d), all the training points are treated as independent and our method in no way attempts to forecast the time evolution of SSH.

**CNN architecture and performance measures**. The optimal CNN architecture used in this study is schematically shown in Fig. 2, consisting of four pooling and three convolutional layers. After the third convolution the features are flattened into a 1D vector and passed into a two-layer fully connect neural network which finally predicts the flux. ReLu (i.e. max(0, x)) was used as the nonlinear activation function since it outperformed the sigmoid and a hyperbolic tangent functions. In summary, a convolutional layer works by convolving a small weight filter matrix (Fig. 2b, rightmost column) over the input image and then passing each output pixel through the activation function. The activation function serves to add non-linear properties to the network, allowing it to learn highly complex mappings. These layers are stacked on top of one another (this is the 'deep' in 'deep learning') in between pooling layers which downsize the image by selecting the maximum of 4 local pixels. The filter weights and fully connected weights are learned (trained) by backpropagtion; the derivative of a cost measuring the error in the flux prediction is found with respect to all of the weights in the network which are then each updated a small amount so as to reduce the loss. The power of CNNs comes from the fact that the filters of each convolutional layer are learned from the data as part of the training process, not specified a priori.

The hyperparameters were chosen to optimize the network for the task of flux reconstruction, specifically: the convolution matrices had horizontal dimensions of $4 \times 4$ and gradient descent was achieved using Kingma and Ba's AdamOptimizer algorithm[61] with default training rate of 0.001. To reduce overfitting, dropout

(where, whilst training, certain units are randomly set to zero, forcing the network to perform a sort of averaging over many possible models, rather that developing complex co-adaptations specific to the dataset it is training on) with a probability of 30% was implemented between the first and second fully connected layers. L1 and L2 weight regularization (essentially penalizing the network for having weights which are too large) was tested for varying strengths over 6 orders of magnitude. L1 regularization had no affect, L2 regularization gave a small increase in performance ( ~ +0.02 skill) with a strength of 0.0001, and further reduced overfitting. The CNN was set to minimize the loss function chosen as the mean squared error between the true flux $\mathbf{y}_t$ and the CNN prediction $\mathbf{y}_p$. The network was coded in Python using Google's machine learning package TensorFlow[72].

To evaluate the performance of the CNN and other data-driven methods, we use the skill, $S$, and the correlation coefficient, $R$, defined as

$$\mathrm{Skill} = 1 - \left( \frac{\frac{1}{N}\sum_{i=1}^{N}(\mathbf{y}_{p,i} - \mathbf{y}_{t,i})^2}{\sigma_{\mathbf{y}_t}^2} \right)^{\frac{1}{2}} \quad \text{and} \quad R^2 = \left( \frac{\frac{1}{N}\sum_{i=1}^{N}(\mathbf{y}_{p,i} - \overline{\mathbf{y}}_p)(\mathbf{y}_{t,i} - \overline{\mathbf{y}}_t)}{\sigma_{\mathbf{y}_p}\sigma_{\mathbf{y}_t}} \right)^2, \tag{6}$$

where $\sigma_{\mathbf{y}_t}, \sigma_{\mathbf{y}_p}$ are the standard deviation of the true and predicted eddy heat fluxes $\mathbf{y}_t, \mathbf{y}_p$. The skill and the correlation coefficient both approach 1 for a perfect prediction; however, there are important differences in the interpretation of these metrics. The skill, a monotonically decreasing function of the squared loss, can be negative if the prediction is worse than the data mean, i.e. predicting the average of the eddy heat fluxes corresponds to a zero skill. The square of the correlation coefficient, $R^2$, provides a useful measure of a fraction of variance that is explained by the prediction, but it in some cases fails to be a reliable measure of accuracy as it is insensitive to shifts in the mean or multiplication by a constant multiple. Throughout the paper we specify both metrics.

Since NN training involves stochasticity in both defining its initial parameters and during their batch optimization, we defined their performance based on an average metrics in the following way. First, we evaluate NN skill on a validation dataset (10 times per epoch) and obtain top 100 results. Second, we evaluate the skill on the entire test data using each of the CNN model parameters corresponding to the top 100 validation skills. The average of the test skill and its standard deviation is the one we report in our study. The choice of using averages over top 100 validation skills biases our skill metrics slightly lower compared to the maximum skill, but the difference is only about 5%.

**Benchmarks and additional tests**. We compare CNN performance to a number of more standard statistical techniques and summarize the results in Fig. 5. Where applicable we explored various architectures/hyperparameters but only report the best result. The methods include:

- *Linear regression.* First we assume $\psi_2 = \psi_1$ then perform simple linear regression on the predicted flux. To first order Fig. 1b shows these two fields are proportional. Given the estimated $\psi_2$ we then calculate the eddy heat flux, which in a non-periodic domain doesn't have to be zero even if $\psi_2$ is proportional to $\psi_1$. 2 trainable parameters, O(1 cpu second).

- *Principal component analysis (PCA).* By finding the PCA basis set for concatenated training-$\psi_1$ & $\psi_2$ snapshots and retaining an optimal number of modes, test-$\psi_1$ images can be used to find an estimate for their corresponding $\psi_2$ field, from which the eddy heat flux is found. PCA is also known under the names of Proper Orthogonal Decomposition or EOFs. Zero trainable parameters, O(1 cpu hour) to find PCs.

- *Support vector machine (SVM).* Regression with a radial basis function kernel. SVMs are an early but effective form of supervised machine learning good at classification and regression. O(1 cpu hour) to train.

- *Random forest regression.* We implement random forest with 75 trees estimators. Another commonly used machine learning algorithm for regression problems. O(1 cpu hour) to train.

- *Fully connected neural networks (FC NN).* We use a basic neural network with 2 hidden layers of 100 and 10 neurons respectively, ReLU activation, mean square error as the loss function, and no dropout. This is a basic form of deep supervised learning which treats the input images as flattened vectors. The results do not significantly change if a higher number of neurons is used. O (400,000) trainable parameters, O(1) cpu hour to train.

- *Convolutional neural networks (CNNs).* These networks have the advantage of explicitly treating the input as images (spatially ordered data) by applying convolutional filters with adjustable parameters. Here we show the best results from VGG-type[73] and ResNet-type[66] architectures. Our VGGNet architectures are of various complexity depending on the number of convolutional filters and number of neurons used in dense layers. The hyperparameters for the CNN architectures referred by the numbers in Fig. 3:

  (1) Two 3×3 convolutional layers (4 filters each) and 2×2 max-pooling layers followed by a fully connected layer.
  (2) A single 4×4 convolutional layer (8 filters) and a single max-pooling (4×4 strides, 4×4 poolsize) with no hidden dense layers.
  (3) Three 4×4 convolutional layers and 2×2 max-pooling layers followed by a hidden layer with 10 neurons.

(4) Three 4×4 convolutional layers (8,16, and 32 filters) and corresponding 2×2 max-pooling layers (4×4 poolsize) followed by a hidden layer with 128 neurons.

(5) Sixteen 4×4 convolutional layers and three 2×2 max-pooling layers followed by a hidden layer with 128 neurons (similar to the VGG16 architecture).

(6) Five 4×4 convolutional layers (8,16,32,64, and 128 filters) and corresponding 2×2 max-pooling layers (4×4 poolsize) followed by a hidden layer with 128 neurons.

(7) Corresponds to the ResNet architecture containing 52 layers (50 convolutional and two dense layers) with skip connections[66]. Dropout of 30% was used on the last fully connected layer in all networks. The network graphs can be downloaded as png-files following the data link in the Data Availability section.

The complexity of these CNNs varies significantly from $O(300)$ trainable parameters and $O(10$ cpu minutes) to train for the smallest CNN to $O(23$ million) trainable parameters taking $O(1000$ cpu hours) to train for ResNet.

A range of additional tests were performed to confirm that a skill of 0.36 is approximately an upper bound, corresponding to a correlation with true heat flux of $R \approx 0.8$ with $R^2 \approx 0.6$. These test included applying filters to reduce the flux contribution from eddies crossing subdomain boundaries, increasing and reducing the QG model resolution, increasing the subdomain size to 2000 km, reducing the subdomain size to 500 km, exploring the CNN architecture by changing the number of filters and fully connected layers, oversampling the eddy flux outliers to obtain a more uniform distribution among the training dataset. Without exceptions, the average test skill obtained over the final 3 epochs of training was never above 0.36, indicating that this may be a dynamically constrained upper bound on the information contained in SSH snapshots.

## Data availability

The necessary procedures to generate the data and reproduce the machine learning techniques have been outlined in the manuscript. The datasets and python scripts used in our study have been published in a Figshare repository[74]. We provided $O(10^5)$ SSH snapshots of mesoscale turbulence and corresponding domain-averaged eddy heat fluxes as simulated by the two-layer QG model and split into training and validation data; the data and Python/TensorFlow scripts including neural network architectures graphs and hyperparameters that reproduce our training results can be downloaded here: https://doi.org/10.6084/m9.figshare.11920905.v1. If additional data is needed, the QG model that was used to generate the samples is available upon request from the authors.

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

## Acknowledgements

The authors gratefully acknowledge support from Charlie Trimble as well as the David and Lucile Packard Foundation. This work was partially completed during Caltech's Summer Undergraduate Research Fellowship Program (SURF), and we thank the SURF staff for their assistance. Glenn Flierl provided the QG turbulence code used in this study. This study benefited from conversations with Frederick Eberhardt and RJ Antonello.

## Author contributions

G.E.M. conceived the study; T.M.G. and G.E.M. performed the research; all authors analyzed the results and contributed to writing of the paper; the research was supervised by A.F.T. and G.E.M.

## Competing interests

The authors declare no competing interests.
