## [Peer Review File · Nature Communications]

Review: Deep learning to infer eddy heat fluxes from sea surface height patterns of mesoscale turbulence

October 10, 2019

The authors investigate the use of deep convolutional neural networks (CNNs) to predict eddy heat fluxes from sea-surface height snapshots in a non-intrusive manner. They utilize an idealized numerical test case to generate training data and assess the validity of their assumptions for different network configurations and against other data-driven techniques. The work is well executed and encourages debate on the real-world utilization of deep learning for geophysical applications. I would like to recommend the publication of this manuscript but after the following points have been satisfactorily addressed

1. My understanding of this article is that the authors have assumed each input image (i.e., the SSH snapshots) is i.i.d in space and time. The space assumption may still hold (looking at the contour plots where there is a significant isotropy in the input features) but I'm not so sure about time. Have the authors considered sequential techniques such as a CNN-LSTM implementation where multiple SSH snapshots are used to predict the eddy heat fluxes. Figure 1e alludes to this where there is autocorrelation between input features (in time). If there are any practical considerations limiting this (which I may be unaware of), these should be included in the paper.
2. Another concern is the fact that structured grid assumptions are intrinsic to what the authors are proposing. I am always a bit wary of CNN results for data that is intrinsically *unstructured* when it comes to practical use - because this implies some sort of interpolation will be required to project data to a structured grid. This interpolation, in the context of high frequency information content, is sure to make effective use of CNNs super challenging. Have the authors considered this, for instance, by sampling in an unstructured fashion from the structured grid? My first instinct, would be to parameterize a relationship that is local in nature, perhaps with input features that take the location of measurement into account.

3. The authors perform a comparison with some other methods and claim that the CNN is superior and claim an upperbound in performance. This is a very strong statement and needs to be reassessed. The authors have not performed a thorough hyperparameter optimization of any of the methods stated there (ideally this needs to include a Bayesian optimization component in the workflow). At the very least, the authors should mention the number of trainable parameters (and offline training duration) for each approach for the purpose of context. The FCNN with 2 layers will have *significantly* fewer trainable parameters than their ResNet with 52 layers.
4. On page 5 the authors claim that 20 epochs, the CNN starts to overfitting. To rectify this they have used dropout regularization, did the authors also look at L_1 techniques to constrain the weights of the network? Or for instance, the Jacobian of the network. This is not a major concern - I am always curious when one technique does better than the other.

I would be happy to review a revised manuscript.

Reviewer #2 (Remarks to the Author):

This study addresses a challenging and important problem of extracting information on 3D oceanic fields from surface variables. Specifically, the study explores whether it is feasible to estimate large-scale heat transport from SSH anomalies, using an example of a highly idealized baroclinic QG flow. The flow consists of a broad mean zonal flow and energetic mesoscale currents ("eddies") generated by the baroclinic instability. Using convolutional neural networks (CNNs), the authors demonstrate that SSH can be used to predict about 64% of the eddy heat flux variance, and the results suggest that this could be an upper limit on the predictive skill. The manuscript is generally well written, the method is promising, and the science is new and interesting. I recommend publication after a revision. My specific comments are listed below.

- "mesoscale eddies – vortices with scales ..." (p.1) -- isolated vortices are only one type of mesoscale currents. Please clarify that the eddy heat transport discussed in this study is induced by all types of mesoscale currents ("eddies"), including multiple fronts, jets, nonlinear waves and filaments.
- "... eddy heat fluxes ..." (p.1) – only the divergent component of the heat flux is important in the dynamics (see Jayne and Marotzke 2002)
- "... maximized when subsurface flows are $\pi/2$ -phase shifted ..." (p.2) – This property is not universal, although it does indeed apply to the 2-layer QG model considered here. Generally, one needs a correlation between temperature and velocity anomalies.
- "From a QG perspective,..."(p.2) – I do not understand the purpose of this sentence. Deep circulation is always associated with PV anomalies, which, in turn, may or may not be associated with the surface flow.
- "Since the mean flow does not change ..." (p.2) – the mean flow, defined using, for example, a spatial average, is significantly modified by eddies via the so-called wave-mean-flow interactions.
- "... mesoscale eddies approximately 200km in size" (p.3) – The mesoscale anomalies seen in Fig.1a have various shapes and sizes, and I do not see why we need to define a typical eddy size.
- Section "Deep CNNs predict eddy heat fluxes from SSH snapshots" (p.5): I could not understand how the CNN method works from the description in the paper, and I suspect that most readers of this manuscript will find themselves in a similar situation. The technique is promising and may not be completely new, but the authors should not assume that the readers are familiar with it. The authors should expand the description of the CNN method, by providing didactic examples and explaining its terminology (e.g., "convolution layers", "neuron layers", "epochs" and "non-linear activation function"). Unfortunately, I did not find Fig.2 very helpful, since many of its elements ("16x16x16", "flatten" and "dropout") are not even explained.
- Fig.2b: Assuming that the left two panels show SSH anomalies, I do not see cyclones (negative SSH anomalies) in these plots. Perhaps I am missing something. I am also wondering how the distinction between cyclones and anticyclones is made, since many of mesoscale anomalies do not have a form of isolated vortices.
- "... characteristic parameters of a mid-latitude baroclinic currents as the Gulf Stream ..." (p.8) – it is unclear why these numbers should be relevant particularly to the Gulf Stream, and not, for example, to the ACC.
- p.9: please state that this study focuses on the meridional heat flux in the upper layer only.
- According to equation 5, the heat flux can be separated into a dynamically significant "coupled flux" and a "trivial" component, and the latter component is then ignored because its domain average is zero. A conventional decomposition of the heat flux into the divergent and rotational components (Jayne and Marotzke (2002) is based on the idea that only the divergent component is important, since the rotational part does not affect temperature. With proper boundary conditions, one can then define a potential function, whose gradient will give the divergent part of the heat flux. It is easy to see that the domain average of the divergent flux over the truly double-periodic domain is then zero. I understand that the domain in this study is not strictly double-periodic, since there is a mean meridional isopycnal slope, but the discussion following the equation 5 is not strictly accurate. Perhaps, the overbar in equation 5 should stand for the zonal average instead of a domain average?
- "... even in the baroclinically-unstable flow ..." – Please clarify this sentence. Although it is true that

growing baroclinic waves are phase-shifted in the vertical, a fully developed mesoscale flow over flat bottom is likely to have a strong barotropic component and may not have a strong vertical shear.

Igor Kamenkovich

Review of “Deep learning to infer eddy heat fluxes from sea surface height patterns of mesoscale turbulence”

Ryan Abernathey

Overview

I read this paper with great interest and excitement. I am hopeful about the role deep learning can play in geophysical fluid dynamics and the problem of ocean heat transport in particular. I agree with the authors’ motivation; inferring these fluxes from remote sensing observations is a useful problem to pursue. I commend them for their innovative work on an important problem. I wish their project to succeed and for their paper to be published eventually in a suitable high impact journal.

However, I believe the study contains a serious flaw. The quantity that the authors have chosen to learn from the QG simulation data, specifically, the meridional heat flux averaged over a 1000 km subdomain, is the wrong target. The quantity that appears in the heat conservation equation is the heat flux *divergence*. The divergence is special because it is frame invariant and coordinate independent. The heat flux vector itself does not uniquely determine the divergence; it is subject to a gauge freedom. In other words, there are infinitely many possible heat fluxes vector fields that have the same divergence. This problem, originally identified by Marshall and Shutts (1981) and recently reviewed by Griesel et al. (2009), has vexed studies of ocean heat fluxes for decades. It cannot be bypassed with deep learning. Furthermore, the authors don’t even learn the full heat flux vector, but only one component of it. My conclusion is that the study should be re-done with the heat flux divergence as the target of learning. I base this recommendation on a survey of the prior literature and on a series of independent calculations I performed when writing this review.

Beyond this crucial point, the study is extremely well written and presented. The hierarchy of machine-learning models is comprehensive and convincing. I will not nit

pick wording or other minor details. My review focuses exclusively on this central issue.

Rotational and Divergent Fluxes

The raw heat flux vector generally contains a large “rotational” component, which makes no contribution to the divergence. Because of this, ocean eddy parameterizations typically seek to parameterize just the divergent part of the flux (Marshall and Shutts, 1981). In model diagnostics, the rotational flux can be eliminated by integrating around any closed contour. A common example is the meridional eddy heat flux (Jayne and Marotzke, 2002), which can be defined as an integral around a closed latitude circle (or a loop including the ocean basin boundary). The heat flux in this sort of QG model is typically analyzed via a global spatial integral (e.g. Thompson and Young, 2007), which similarly eliminates the rotational component. However, in this paper, the eddy heat flux is defined as the meridional projection of the local heat flux vector, averaged over an open subdomain. All evidence from prior literature suggests that this quantity should be dominated by the rotational component, which has no impact on the physics. As stated by Griesel et al. (2009), in the analysis of a realistic high-resolution ocean simulation:

The divergent component accounts for no more than 5% of the total eddy heat transport, decreasing towards zero with increasing lengthscale. Hence, the rotational component dominates all lengthscales and cannot be eliminated simply by averaging over any length scale.

But averaging over a length scale (specifically, 1000 km), is precisely what was done in this paper.

Numerical Experiments

I assume the authors will be reluctant to take my suggestion, since it requires re-training of all their models. To make the case for why it is important, I reproduced their simulation scenario using the open-source model pyqg and performed my own analysis¹. I used identical parameters to the ones stated in their manuscript. As shown in Fig. 1, the upper-layer PV field is qualitatively identical to the one shown in their Fig. 1.

¹Code available at <https://nbviewer.jupyter.org/gist/rabernat/471f0e9662b3d77f65774e972377b04c>

Figure 1: Upper layer PV from my simulation. All parameters were chosen identical to the simulations in the manuscript. The sub-region analyzed is indicated by a purple square.

Figure 2: Heat flux vectors and heat flux convergence. Left panel shows the full heat flux, with the convergence in blue / red. Upper layer ψ contours are overlaid. The Right panel shows the same, but showing only the divergent heat flux vectors. The arrow scale is 1/5 that of the full heat flux. Both vector fields have the same divergence!

The “raw” heat flux vector HF is defined as

$$HF = \mathbf{v}_q h_1 \quad (1)$$

$$= (-h_1 \partial_y \psi_1, h_1 \partial_x \psi_1) \quad (2)$$

The heat flux convergence HFC is given by

$$HFC = -\nabla \cdot (\mathbf{v}_q h_1) \quad (3)$$

$$= \partial_x (h_1 \partial_y \psi_1) - \partial_y (h_1 \partial_x \psi_1) \quad (4)$$

We can isolate the rotational and divergent parts via a Helmholtz decomposition:

$$HF = -\nabla \phi + \nabla \times \hat{k} \Psi . \quad (5)$$

The first term is the divergent part, while the second is the rotational. Note that

$$HFC = \nabla^2 \phi . \quad (6)$$

The rotational part does not enter.

Normally the Helmholtz decomposition is ambiguous due to an extra degree of freedom in the boundary conditions (Fox-Kemper et al., 2003). Fortunately, in a doubly periodic model such as this one, there are no boundary conditions, and there is an exact solution. Given a diagnosed *HFC*, the Laplacian in (6) is trivially inverted in Fourier space:

$$\hat{\phi} = -K^{-2}\widehat{HFC} \quad (7)$$

where $\hat{\cdot}$ indicates the Fourier transform and $K^2 = k^2 + \ell^2$. From this, we can calculate the individual components of the divergent flux

$$\widehat{HF}_{div} = (-ik\hat{\phi}, -i\ell\hat{\phi})$$

and transform back to real space.

The raw heat flux and the divergent part from a single snapshot are both shown in Fig. 2. The overall magnitude of the raw flux is roughly 5 times larger. They appear qualitatively very different; the full flux is dominated by the rotational component circulating around eddy-like features (in agreement with past studies, i.e. Marshall and Shutts, 1981; Griesel et al., 2009), while the divergent part points from regions of divergence to regions of convergence. *Both fields have the exact same divergence!*

We now examine timeseries of the heat flux convergence, as well as the components of both the raw and divergence heat flux vectors, all averaged over a 1000 x 1000 km box. The average value of the heat flux convergence appears in the prognostic equation for the heat content, representing the net heating of this region due to advection; it must be balanced either by a tendency in heat content or by an external forcing (e.g. air-sea heat flux). In contrast, the vector components averaged spatially have no clear physical interpretation, as they do not appear in any prognostic equation. These timeseries are shown in Fig. 3.

The immediate visual impression is that none of the vector components is correlated with the divergence. This is confirmed by calculating the correlations; The highest correlation (0.19) is obtained between the HFC and the divergent meridional heat flux component. The correlation between HFC and the raw meridional heat flux component (the target of learning in the manuscript) is 0.02. *Thus, even if one could predict the raw heat flux perfectly, it would be useless for evaluating the heat budget.* This is the foundation of my argument that the paper is flawed.

Summary

I feel strongly that the paper should not be published in its current form. By publishing the dataset and encouraging machine-learning experts to tackle the problem,

Figure 3: Heat flux timeseries, averaged over 1000 x 1000 km box. Upper panel shows heat flux convergence. Lower panel shows individual components of the raw and divergent heat flux vectors.

the authors are effectively trying to open this challenge to a broad community. But, for the reasons stated above, it is the wrong problem to attack. We therefore risk diverting significant scholarly energy down a sub-optimal path.

The best examples of machine learning applied to physics have used physical insights, conservation principles, and invariances to guide the input features and loss functions. For instance, the now classic paper of Ling et al. (2016) invoked the frame invariance of the Reynolds stress tensor to formulate a novel “tensor basis” deep neural network. Rather than learning the raw components of the stress tensor, they learned its components in a transformed basis which ensured the resulting constitutive relation was invariant to coordinate transformations. This insight led to a breakthrough; where others had failed to learn a neural-network RANS model from DNS data, they succeeded. A similar opportunity may be possible here. It seems at least plausible that, in re-focusing their efforts on a frame-invariant, coordinate-independent quantity (the heat flux convergence), the authors will surpass the skill limit of 0.36 in their current configuration. This is indeed my hope!

Acceptable responses to my review would be

- Redoing the paper with HFC as the target for learning
- Eliminating the sub-region averaging and redoing the experiment for the whole domain (global average eliminates rotational flux)

- Making a convincing and novel argument why the spatially averaged raw meridional heat flux is in fact a suitable target for machine-learning inference. For example, demonstrating how this quantity appears in a heat budget or other prognostic conservation equation.

Despite this serious critique, I repeat my words of encouragement. I believe this can be an important paper, and I will be happy to see it published eventually. I hope the authors will take my recommendations in the constructive spirit with which they were offered.

References

- Fox-Kemper, B., Ferrari, R., and Pedlosky, J. (2003). On the indeterminacy of rotational and divergent eddy fluxes. *Journal of Physical Oceanography*, 33(2):478–483.
- Griesel, A., Gille, S. T., Sprintall, J., McClean, J. L., and Maltrud, M. E. (2009). Assessing eddy heat flux and its parameterization: A wavenumber perspective from a $1/10^{\text{deg}}$ ocean simulation. *Ocean Modelling*, 29(4):248–260.
- Jayne, S. R. and Marotzke, J. (2002). The oceanic eddy heat transport. *Journal of Physical Oceanography*, 32(12):3328–3345.
- Ling, J., Kurzawski, A., and Templeton, J. (2016). Reynolds averaged turbulence modelling using deep neural networks with embedded invariance. *Journal of Fluid Mechanics*, 807:155–166.
- Marshall, J. and Shutts, G. (1981). A note on rotational and divergent eddy fluxes. *Journal of Physical Oceanography*, 11(12):1677–1680.
- Thompson, A. F. and Young, W. R. (2007). Two-layer baroclinic eddy heat fluxes: Zonal flows and energy balance. *Journal of the Atmospheric Sciences*, 64(9):3214–3231.

Response to Reviewers - Nature Communications Submission NCOMMS-19-539539
“Deep learning to infer eddy heat fluxes from sea surface height patterns of mesoscale turbulence”

T George, G Manucharyan & A Thompson

We would like to thank all the reviewers for the time taken to review our manuscript and for their insightful comments that have led to improvements in the clarity of our manuscript. Below, we address each of the reviewer comments shown in blue with our responses shown in black. While the critical conclusions of our manuscript remained the same, in the updated manuscript we have included several crucial clarification points and have made our statements more precise, emphasizing not only the importance of our idealized study but also its caveats. We have also made minor updates to the figures as specified below.

Response to reviewer 1:

The authors investigate the use of deep convolutional neural networks (CNNs) to predict eddy heat fluxes from sea-surface height snapshots in a non-intrusive manner. They utilize an idealized numerical test case to generate training data and assess the validity of their assumptions for different network configurations and against other data-driven techniques. The work is well executed and encourages debate on the real-world utilization of deep learning for geophysical applications. I would like to recommend the publication of this manuscript but after the following points have been satisfactorily addressed

1. My understanding of this article is that the authors have assumed each input image (i.e., the SSH snapshots) is i.i.d in space and time. The space assumption may still hold (looking at the contour plots where there is a significant isotropy in the input features) but I'm not so sure about time. Have the authors considered sequential techniques such as a CNN-LSTM implementation where multiple SSH snapshots are used to predict the eddy heat fluxes. Figure 1e alludes to this where there is autocorrelation between input features (in time). If there are any practical considerations limiting this (which I may be unaware of), these should be included in the paper.

The reviewer is correct, snapshots are not necessarily i.i.d in time. They decorrelate over a time scale of ~20 days (fig 1e) but are sampled in the training data every 10 days. This is done in order to be in line with what real altimetry data available would provide. A small amount of temporal correlation between training samples should not be an issue for training. It would be interesting to investigate the possibility of using a recurrent neural network (e.g. CNN-LSTM) model to predict the evolution of heat fluxes through time in a modeling study. However, existing satellite altimeters have return periods of the order of a month and rely on crude time-interpolation techniques to provide daily snapshots and hence we expect the information contained specifically in time-series of SSH to be severely contaminated by the interpolation technique so as to make practical implementation of this technique significantly less feasible than CNN estimation. Temporal information is indeed an important issue to explore but because of the additional complications arising due to interpolation of the original data, we believe this is better suited for a separate study as it would require substantial investigations in order to make

accurate and practically-relevant conclusions. We have begun very preliminary investigations into LSTM applications to this problem, please see our comment below.

2. Another concern is the fact that structured grid assumptions are intrinsic to what the authors are proposing. I am always a bit wary of CNN results for data that is intrinsically unstructured when it comes to practical use - because this implies some sort of interpolation will be required to project data to a structured grid. This interpolation, in the context of high frequency information content, is sure to make effective use of CNNs super challenging. Have the authors considered this, for instance, by sampling in an unstructured fashion from the structured grid? My first instinct, would be to parameterize a relationship that is local in nature, perhaps with input features that take the location of measurement into account.

Indeed, existing satellites collect information along their tracks which is later interpolated to provide a view of the ocean on a structured grid and this could reduce the heat flux prediction due to the errors introduced in interpolation process. However, our first altimeter satellite that would provide wide-swath SSH snapshots (in addition to along-track measurements) will be launched about one year from now (NASA SWOT) - its observations could be directly used for heat flux reconstruction as they will come in the form of (structured) 2D SSH snapshots and so would not require spatial interpolation. Nonetheless, we have started work on constructing neural network architectures that operate on ungridded along-track SSH observations as, like you, we also agree that bypassing the gridding step could lead to improved predictions. These architectures utilize 1D convolutions and LSTMs and end up being fundamentally different from our CNNs to the point that adding conclusive results to this manuscript would end up extending it very substantially. Our preliminary study on quantifying the degree to which it is more useful to use the raw unstructured data, as opposed to interpolated data, remains inconclusive because of conflicting processes: while unstructured data does not contain interpolation errors, it provides a much more difficult task for neural network training as we cannot make such easy use of CNNs for simplified information extraction from spatial patterns. Because of additional complications that arise when using these fundamentally different neural network architectures, we judge that the exploration of neural network performance on unstructured datasets will not substantially increase the value of this manuscript but it would certainly add a substantial amount of new material, and hence it will be better to publish it in a separate follow-up study.

3. The authors perform a comparison with some other methods and claim that the CNN is superior and claim an upperbound in performance. This is a very strong statement and needs to be reassessed. The authors have not performed a thorough hyperparameter optimization of any of the methods stated there (ideally this needs to include a Bayesian optimization component in the workflow). At the very least, the authors should mention the number of trainable parameters (and offline training duration) for each approach for the purpose of context. The FCNN with 2 layers will have significantly fewer trainable parameters than their ResNet with 52 layers.

We are not claiming that we have found an upper bound, rather we are stating that a theoretical upper bound may well exist given that one has no grounds to assume perfect performance ($\text{skill} = 1$) could ever be achieved since we are dealing with intrinsically partial observations of the system. We stress that it is important to quantify what that bound might be because studies might make ungrounded

assumptions that the upper bound is 1 and attempt to overfit the results to reach it. We, however, clearly state in the paper that our method is not *proving* that we have found an upper bound, but instead we are demonstrating that CNNs could already achieve a skill=0.36 and we hope that a broader deep learning community would find interest in exploring architectures that could improve on this. We have reiterated this statement in our conclusions.

Thank you for the advice on exploring hyperparameters: we have indeed explored a large number of hyperparameters for each method but we have reported only the best results in each case (i.e. figure 4). We did not, as you pointed out, perform a Bayesian optimization of the CNN hyperparameters however, instead, we did perform a standard hyperparameter sweep (figure 3f) over the CNN architecture (depth and total number of parameters varied). Loss functions, gradient descent algorithms and weight regularisations (see point 4 below) were also considered and this has been stated in line 143 of the manuscript. We believe this is sufficient to support our conclusions. Finally, as you suggested, we have updated the manuscript to include the number of trainable parameters and offline training time for each technique (see Benchmarks and additional tests section).

4. On page 5 the authors claim that 20 epochs, the CNN starts to overfitting. To rectify this they have used dropout regularization, did the authors also look at L1 techniques to constrain the weights of the network? Or for instance, the Jacobian of the network. This is not a major concern - I am always curious when one technique does better than the other.

In response to this comment please see figures below where we have tested the addition of L1 and L2 regularization constraints on the weights, varying the strength of this regularization over 6 orders of magnitude. L1 regularization does not appear to have a significant effect. L2 regularization, on the other hand, if tuned correctly, can give a small increase in performance. We thank the reviewer for the advice that led to this finding and will update the manuscript accordingly to include this new 'best' result.

I would be happy to review a revised manuscript.

Response to reviewer 2, Igor Kamenkovich:

This study addresses a challenging and important problem of extracting information on 3D oceanic fields from surface variables. Specifically, the study explores whether it is feasible to estimate large-scale heat transport from SSH anomalies, using an example of a highly idealized baroclinic QG flow. The flow consists of a broad mean zonal flow and energetic mesoscale currents (“eddies”) generated by the baroclinic instability. Using convolutional neural networks (CNNs), the authors demonstrate that SSH can be used to predict about 64% of the eddy heat flux variance, and the results suggest that this could be an upper limit on the predictive skill. The manuscript is generally well written, the method is promising, and the science is new and interesting. I recommend publication after a revision. My specific comments are listed below.

- “mesoscale eddies – vortices with scales ...” (p.1) -- isolated vortices are only one type of mesoscale currents. Please clarify that the eddy heat transport discussed in this study is induced by all types of mesoscale currents (“eddies”), including multiple fronts, jets, nonlinear waves and filaments.

We have appended the sentence following this with: “... though other types of mesoscale currents such as jets or fronts can, in regimes not covered here, can also dominate.”. The heat transport discussed in this study does not extend to these flows though they are of course valid and important in other, less ubiquitous, flow regimes.

- “... eddy heat fluxes ...” (p.1) – only the divergent component of the heat flux is important in the dynamics (see Jayne and Marotzke 2002)

We have adjusted introduction paragraph 1 to mention, as you say, the relevance of the heat flux divergence, for example, in the ocean heat budget equation. We have also added the following sentence into the methods:

“It is well known that the total heat flux can be partitioned into divergent and rotational components (as opposed to our ‘coupled’ and ‘trivial’ components), of which only the divergent component contributes to the heat flux divergence \cite{foxkemper2013}. Here we reconstruct the total heat flux - which amounts to predicting the coupled component - from which the heat flux divergence (or any other quantity) can be determined. Note it is only the heat flux divergence that contributes to a local heat flux tendency.”

- “... maximized when subsurface flows are $\pi/2$ -phase shifted ...” (p.2) – This property is not universal, although it does indeed apply to the 2-layer QG model considered here. Generally, one needs a correlation between temperature and velocity anomalies.

Clarified

- “From a QG perspective,...”(p.2) – I do not understand the purpose of this sentence. Deep circulation is always associated with PV anomalies, which, in turn, may or may not be associated with the surface flow.

After some discussion we have decided to leave this sentence in place as we believe it is crucial for the motivation as to why we use a QG modal. Essentially it sets up the sentence which follows it. Deep circulation is not always associated with PV anomalies in the deep ocean, instead one could have PV anomalies localized at the surface that cause deep ocean currents. However, for the existence of eddy heat fluxes in baroclinically unstable processes the PV anomalies are required to be in *both* layers, which makes our problem non-trivial because knowing only surface the streamfunction it is insufficient to straightforwardly reconstruct PV anomalies in either of the layer .

The clarified text now appears in the following way: “From a quasigeostrophic (QG) perspective (i.e. large-scale, slowly-evolving turbulence in rotationally-constrained fluids), eddy heat fluxes emerge from baroclinic instabilities for which the unobserved bottom flows are affecting and are affected by the observed surface flows. Thus, the observed eddy patterns in an SSH snapshot might constrain the posterior distribution of subsurface flow and thus contain at least partial information about the corresponding eddy fluxes.”

- “Since the mean flow does not change ...” (p.2) – the mean flow, defined using, for example, a spatial average, is significantly modified by eddies via the so-called wave-mean-flow interactions.

We have clarified this to “Since the background mean flow in our model does not change ...”

- “... mesoscale eddies approximately 200km in size” (p.3) – The mesoscale anomalies seen in Fig.1a have various shapes and sizes, and I do not see why we need to define a typical eddy size.

We agree and have removed reference to this size scale.

- Section “Deep CNNs predict eddy heat fluxes from SSH snapshots” (p.5): I could not understand how the CNN method works from the description in the paper, and I suspect that most readers of this manuscript will find themselves in a similar situation. The technique is promising and may not be completely new, but the authors should not assume that the readers are familiar with it. The authors should expand the description of the CNN method, by providing didactic examples and explaining its terminology (e.g., “convolution layers”, “neuron layers”, “epochs” and “non-linear activation function”). Unfortunately, I did not find Fig.2 very helpful, since many of its elements (“16x16x16”, “flatten” and “dropout”) are not even explained.

We have expanded the CNN methods section as you suggested. The “16x16x16” refers to the size of the object in the given hidden layer (16 by 16 pixels, by 16 features, colour coordinated with the arrows they sit next to), something we have now clarified in the figure caption. Flatten and dropout have now been defined in methods too.

- Fig.2b: Assuming that the left two panels show SSH anomalies, I do not see cyclones (negative SSH anomalies) in these plots. Perhaps I am missing something. I am also wondering how the distinction between cyclones and anticyclones is made, since many of mesoscale anomalies do not have a form of isolated vortices.

The colours in figure 2b no longer show SSH anomalies but, rather, the activation of the first hidden layer of neurons. The activation function used ($\text{ReLU}(x) = \max(0,x)$) rectifies the signal so that the

cyclonic features picked out by the network (orange triangle in upper panel of 2b corresponds with orange triangle in 2a) are represented with positive (yellow) activations.

We agree that, where the mesoscale anomalies do not form isolated vortices, it is meaningless to define cyclones over anticyclones. The distinction is only meaningful in the case where the eddies do appear as isolated roughly circular vortices.

- “... characteristic parameters of a mid-latitude baroclinic currents as the Gulf Stream ...” (p.8) – it is unclear why these numbers should be relevant particularly to the Gulf Stream, and not, for example, to the ACC.

In the manuscript we said “...such as the Gulf Stream” but have replaced this with “...such as the Gulf Stream or the Antarctic Circumpolar Current”.

- p.9: please state that this study focuses on the meridional heat flux in the upper layer only.

Done.

- According to equation 5, the heat flux can be separated into a dynamically significant “coupled flux” and a “trivial” component, and the latter component is then ignored because its domain average is zero. A conventional decomposition of the heat flux into the divergent and rotational components (Jayne and Marotzke (2002) is based on the idea that only the divergent component is important, since the rotational part does not affect temperature. With proper boundary conditions, one can then define a potential function, whose gradient will give the divergent part of the heat flux. It is easy to see that the domain average of the divergent flux over the truly double-periodic domain is then zero. I understand that the domain in this study is not strictly double-periodic, since there is a mean meridional isopycnal slope, but the discussion following the equation 5 is not strictly accurate. Perhaps, the overbar in equation 5 should stand for the zonal average instead of a domain average?

We decompose the flux in this way as it best clarifies the mathematically challenging component of what we are estimating. We did check the ability of a CNN to calculate the ‘trivial’ component and it does so perfectly with ease, as one would expect. Rotational and divergent decomposition is equally valid. With regards to your latter point, the ‘trivial’ component is non-zero in our subdomains since they are not doubly periodic. The point is that it is (i) ‘trivial’ to calculate given SSH and (ii) only dependent on the boundaries of the subdomain, not intrinsic to the processes going on within the subdomain. If we move our subdomain ~200km in any direction this trivial term will change significantly as a totally different arrangement of eddies will now be crossing the boundary. Conversely, the coupled term will not change nearly so much.

The take home here is that, once we know the full heat flux (i.e. trivial plus coupled) one is free to then calculate whichever component of this they desire for future requirements for example the divergent component which, as you stated, is the only part which affects temperature. Overbar refers to averaging over subdomain (stated in paragraph above eq 5) which I believe is what you mean by ‘zonal averaging’.

Finally, and to be rigorous, we pointed out in the paper that for the purposes of heat budget estimation only the divergent component of the eddy heat flux provides a contribution. Calculating the heat flux divergence or decomposing the flux into divergent and rotational components (i.e. implementing the

Helmholtz decomposition) can be done after the sub-domain averaged heat flux has been estimated using our deep learning techniques. This is because the divergence or Helmholtz decomposition operators commute with the procedure of calculating sub-domain averages.

- “... even in the baroclinically-unstable flow ...” – Please clarify this sentence. Although it is true that growing baroclinic waves are phase-shifted in the vertical, a fully developed mesoscale flow over flat bottom is likely to have a strong barotropic component and may not have a strong vertical shear.

As this sentence is not central to our line of reasoning we have removed it to avoid confusion.

Response to reviewer 3, Ryan Abernathy:

Overview

I read this paper with great interest and excitement. I am hopeful about the role deep learning can play in geophysical fluid dynamics and the problem of ocean heat transport in particular. I agree with the authors’ motivation; inferring these fluxes from remote sensing observations is a useful problem to pursue. I commend them for their innovative work on an important problem. I wish their project to succeed and for their paper to be published eventually in a suitable high impact journal.

However, I believe the study contains a serious flaw. The quantity that the authors have chosen to learn from the QG simulation data, specifically, the meridional heat flux averaged over a 1000 km subdomain, is the wrong target. The quantity that appears in the heat conservation equation is the heat flux divergence. The divergence is special because it is frame invariant and coordinate independent. The heat flux vector itself does not uniquely determine the divergence; it is subject to a gauge freedom. In other words, there are infinitely many possible heat fluxes vector fields that have the same divergence. This problem, originally identified by Marshall and Shutts (1981) and recently reviewed by Griesel et al. (2009), has vexed studies of ocean heat fluxes for decades. It cannot be bypassed with deep learning. Furthermore, the authors don’t even learn the full heat flux vector, but only one component of it. My conclusion is that the study should be re-done with the heat flux divergence as the target of learning. I base this recommendation on a survey of the prior literature and on a series of independent calculations I performed when writing this review.

Beyond this crucial point, the study is extremely well written and presented. The hierarchy of machine-learning models is comprehensive and convincing. I will not nit pick wording or other minor details. My review focuses exclusively on this central issue.

A PDF with figures and additional discussion followed this overview which we have not included here. Review 3’s summary is pasted below

Author Response:

Firstly, thank you for this careful and thorough review which you have clearly put a lot of effort into. We have read it carefully and taken your points on board. In summary, you have asked us to redo the study with the heat

flux convergence ($\text{HFC} = -\nabla \cdot \mathbf{HF}$) as the learning target. We would like to respond by presenting an argument for why this is not in fact required and why the sub-domain averaged raw meridional heat flux is the correct quantity to search for.

We believe your claim “*The heat flux vector itself does not uniquely determine the divergence; it is subject to a gauge freedom*” on page 1 is incorrect. Rather, it is the other way around; the divergence does not uniquely determine the heat flux. We are not nit-picking here, it is an important point since it implies, if you know the full flux, you can calculate the divergence. And the full flux is exactly what we endeavour to find, in two components: the coupled component (found using deep learning, this is the focus of our paper) plus the trivial component (which is a unique and known function of SSH), both defined as per equation 5 in the manuscript. In essence, our counter response can be summarised by saying that once you have an estimate of the full flux (by CNN or otherwise) it is yours to do with as you wish, including finding the HFC which can be used downstream, for example, in the prognostic equation for ocean heat content. If we immediately aimed to find the convergence, we would then lose the ability to go back and find the flux, due to this gauge freedom.

You present concerns with the spatial averaging procedure. Here we clarify that the operation of averaging the flux field over subdomains (here done by convolving with a 1000 x 1000km top-hat) commutes with the operation of finding the divergent component of the flux field. To demonstrate this we downloaded and ran your pyqg simulation. The figure below is essentially equivalent to figure 2 in your review (i.e. a vector field of the relevant flux component overlaid onto a heatmap of the HFC and contours of the upper layer streamfunction) and shows the heat flux broken into the divergent and rotational components:

Fig 1: The full heat flux is shown in the same manner as Abernathy comments, fig 2. It is split into a purely rotational and purely divergent component. Here we plot the 2000 x 2000km region (effectively 4 subdomains as defined in the paper) in the center of the full 4000 x 4000km domain. As one can see, by the similarity of the leftmost and rightmost fields, the flux is dominated by the rotational component (as stated by Abernathy).

To find the subdomain averaged divergent component we can either:

- Average the flux over subdomains *then* find the divergent component
- Or the other way around: Find the divergent component *then* average over subdomains

In figure 2 we do both of these and show they amount to the same flux:

Fig 2: The divergent component of the heat flux, averaged over sub-domains, can be found in one of two ways: either the full heat flux is averaged over the subdomains, then the divergent component of this is found (right) or the divergent component can first be found and then this field can be averaged over subdomains (left). As we demonstrate above, they are equal. This demonstrates the commutability of the averaging and the divergence operators. Here, averaging is done by convolving with a square top hat of size 1000km x 1000km (i.e. exactly equal to the subdomains we refer to in the paper).

In conclusion: the operations of spatial-averaging and finding divergent components commute, therefore given the subdomain average HF we can then find the subdomain averaged HFC, a quantity required in the heat content equation.

In figure 3 you show the time series of the components of the heat flux and the timeseries of the HFC. Whilst we agree that none of the flux components show any significant correlation with the HFC (which should not be required anyway since $HFC = \text{div HF}$ and hence the time series of HC and HFC are not required to be correlated), we do not believe this renders HF being “*useless for evaluating the [HFC]*” because one could uniquely calculate HFC from the spatial distribution of HF by simply calculating its divergence, using the procedure you described in your review.

As an illustrative example, consider an ocean with a localized mean flow in the zonal direction (East to West), such that there is no mean flow to the North or South of it. Here, the neural network-based heat fluxes could be computed based on eddy patterns in a subdomain covering the core of the zonal current (strong HF) and in another two subdomains to the North and South of the current (weak HF). Given these heat fluxes one could uniquely determine the associated HFC. It is important to also note here that the obtained weak HFC in Fig 2 (or reviewer’s Fig 3) arises because the data comes from a QG simulation that is doubly-periodic and hence its zonal mean heat flux in the direction perpendicular to the mean flow must be uniform, i.e. non-divergent. It is, however, possible in this simulation to have non-zero HFC over some small subdomains that do not entail the whole zonal mean, but that HFC would be localized around individual eddies (Fig. 2) and does not present information related to the intensity of baroclinic instability and the transfer of energy from the available

potential energy of the mean flow -- that information is contained in the zonal mean HF itself. The reason for why we have used the double periodic simulation is that we consider that a subdomain over which the HF is averaged will entail a region of multiple eddies formed over a patch of roughly uniform mean flow, and that in neighboring subdomains, the mean flow could be different and hence the eddy configuration and corresponding HF would also vary spatially, implying that there is a significant HFC.

As a more simple argument for why the lack of correlation in reviewer fig 3 should not be of concern; $\sin(x)$ and $\cos(x)$ have zero correlation over the range $[0, 2\pi]$ but they are related (trivially) by only a phase shift. Given the blue (hf_x) and orange (hf_y) lines in your figure 3 for nearby subdomains, and using the commutativity argument presented above, one can precisely find the heat flux convergence, by the procedure you outlined in your review. The same is true for the red and green curves.

We have updated the manuscript to clarify that, in typical scenarios, it is indeed the HFC which is useful to know (these additions can be found in the introduction and the methods), however we strongly maintain that identifying the subdomain averaged heat flux from SSH observations is a dynamically correct way to formulate a problem.

Reviewer 3 Summary

I feel strongly that the paper should not be published in its current form. By publishing the dataset and encouraging machine-learning experts to tackle the problem, the authors are effectively trying to open this challenge to a broad community. But, for the reasons stated above, it is the wrong problem to attack. We therefore risk diverting significant scholarly energy down a sub-optimal path.

We have demonstrated that the heat flux convergence could be uniquely calculated from the HF itself, so our judgement is that it is not necessary to redo our study using the HFC.

The best examples of machine learning applied to physics have used physical insights, conservation principles, and invariances to guide the input features and loss functions. For instance, the now classic paper of Ling et al. (2016) invoked the frame invariance of the Reynolds stress tensor to formulate a novel "tensor basis" deep neural network. Rather than learning the raw components of the stress tensor, they learned its components in a transformed basis which ensured the resulting constitutive relation was invariant to coordinate transformations. This insight led to a breakthrough; where others had failed to learn a neural-network RANS model from DNS data, they succeeded. A similar opportunity may be possible here. It seems at least plausible that, in re-focusing their efforts on a frame-invariant, coordinate-independent quantity (the heat flux convergence), the authors will surpass the skill limit of 0.36 in their current configuration. This is indeed my hope!

Using frame-invariance in the input/output is potentially a great idea, however, it is not the only one and there are many other ideas that one could think about to surpass the limit that we achieved with our network architecture. It is this kind of response that we hope to get from the scientific community, *i.e.* identify various

approaches to tackle this problem. We have identified and isolated the critical issue with geostrophic turbulence that would be present regardless of whether we consider reconstructing HF or its divergence, the fundamental issue being that SSH contains only partial information about the baroclinically unstable system -- identifying exactly how much useful information is contained in SSH snapshots of eddy patterns is the very topic that we would like the scientific community to address and, from that perspective, we are not restricting anyone from reconstructing the HFC instead of the HF.

Acceptable responses to my review would be

- Redoing the paper with HFC as the target for learning

We feel strongly that this option is not necessary since we have demonstrated above that the subdomain-averaged HFC is equivalent to the divergence of the subdomain-averaged HF.

- Eliminating the sub-region averaging and redoing the experiment for the whole domain (global average eliminates rotational flux)

We have indeed conducted this experiment: estimated the heat fluxes averaged over an entire 1000km doubly-periodic domain and have found a similar range of skills depending on the neural network architecture and the amount of data used. In fact, we started from these types of calculations, but later decided to present our work in terms of sub-domain averages (i.e. 1000km subdomains of a 4000km doubly-periodic domain). Our rationale was that the HF reconstruction skill did not decrease substantially, whereas subdomains would, in practice, allow us to address this problem in a real ocean that is not doubly-periodic. We attach the figure below from the early stages of our research. Here, the domain (right: 1000km x 1000km) is doubly-periodic and the CNN predicts the domain-averaged upper-layer PV flux (which is proportional to the eddy HF due to double periodicity). The network successfully reconstructs the flux (left subfigure) up to a maximum skill of 0.39, which is not significantly different from reconstructing HF in non-periodic sub-domains. We have now mentioned in the manuscript that CNNs can also learn to estimate the heat fluxes in double-periodic domains, as you enquired.

- Making a convincing and novel argument why the spatially averaged raw meridional heat flux is in fact a suitable target for machine-learning inference. For example, demonstrating how this quantity appears in a heat budget or other prognostic conservation equation.

Indeed, if one considers a heat budget, then it is the heat flux convergence that enters the problem and there could not be any other arguments here. What we argue is that HFC can be reconstructed from the total heat flux. From the perspective of ocean energetics, the term with potential energy extraction from the mean flow is expressed in the form of a flux, not the flux divergence. Hence, at least from an energetics perspective, the eddy heat flux itself is a useful quantity. Also, note that the eddy heat flux does not contain the issue of moving reference frames because the quantities in the eddy heat flux are already the perturbations from zonal mean; it could however contain certain other dynamically-imposed symmetries (depending in the nature of the mean flow, i.e. homogeneous or isotropic) that could be enforced during the neural network training.

Despite this serious critique, I repeat my words of encouragement. I believe this can be an important paper, and I will be happy to see it published eventually. I hope the authors will take my recommendations in the constructive spirit with which they were offered.

Once again, thank you for your constructive criticisms and words of encouragement. We are very appreciative of the time you put into this review and, in the process of constructing our response, we learnt a lot and confirmed some important intuitions (commutativity of spatial-averaging and divergence-taking, for example).

REVIEWER COMMENTS

Reviewer #1 (Remarks to the Author):

Thank you for addressing my concerns in your revised manuscript. I believe it is now suitable for publication. Thank you, once again, for making your dataset and code public - this is crucial for a community-driven investigation into scientific machine learning for the geophysical community.

Reviewer #2 (Remarks to the Author):

The authors adequately addressed all my specific comments.
I recommend publication after a minor revision.

My remaining criticism of the paper is the focus on the total eddy heat flux. The main argument is that once the full flux is known, one can perform the Helmholtz decomposition and extract the divergent part. There are two problems with this logic. Firstly, extraction of the divergent flux component is not unique, because it strongly depends on the lateral boundary conditions for the divergent component, which are not uniquely defined. Another problem is that the total eddy heat flux tends to be dominated by the rotational component, and even a relatively small error in the total flux can lead to large errors in the divergent part.

On the other hand, the paper describes a promising method of using SSH to diagnose heat flux – a nonlinear quantity that depends on the 3D flow and temperature structure. One can argue that the degree to which the quantity itself (full heat flux) is critical for describing the temperature evolution is of secondary importance here.

I recommend that the authors move the discussion of the importance of the divergent heat flux ("It is well known that ...") from the end of the Methods section to either the Introduction or Discussion, and expand it, clearly stating that although the results may not lead to accurate estimates of instantaneous heat flux divergence, the main point of the paper is to demonstrate that the method works for using SSH to estimate a nonlinear quantity that depends on 3D velocity and temperature. A direct comparison between the CNN-estimated and actual flux divergence would, of course, be even more convincing, but I expect the corresponding biases to be large.

Igor Kamenkovich

Reviewer #4 (Remarks to the Author):

1) I agree that it is the divergence of the eddy heat flux that should have been considered in the first place. I don't think we are mainly interested in how well the variance of the eddy heat flux can be predicted (e.g. the "wiggles" of alternating large and small heat fluxes), but we want to know what is the net effect, i.e. how much heat is fluxed in and out of a region.

You say that once you have the heat flux, then the divergence can always be computed, but if you have the divergence, then the heat flux cannot be uniquely recovered, which is of course true. The question is are your algorithms "conservative" in the sense that they provide the eddy heat flux that will give you the right divergence? As Fig. 3 shows, while it's true that the "wiggles" in time are quite nicely reproduced by the pattern recognizing algorithms, the magnitude of the heat flux, i.e. the net heat flux over the subdomains, is not. Unless I misunderstood something, in the supplementary material you provide calculations saying that the divergence of the averaged eddy heat fluxes equals the averaged eddy heat flux divergence. But that is trivial.

You should compare the divergence of the predicted eddy heat flux with the divergence of the true eddy heat flux. What would the time series of Fig. 3d look like if you compared the predicted and true divergence? What does the bias in the magnitude of the eddy heat flux do to the divergence of the eddy heat flux? A related question is how does this bias depend on the size of the subdomain? The eddy heat flux locally is very noisy, very large values are next to very small values, the subdomain-averaged eddy heat flux for a smaller subdomain is very different from the subdomain average over a larger domain. All of these questions can be avoided by predicting the divergence of the eddy heat flux in one subdomain.

2) I still think that the CNN method is not explained well-enough for an audience that does not know much or anything about "deep learning" algorithms. The authors should make another effort to explain the principle, before delving into Fig. 2. Also, Fig. 2 is still full of terms that mean nothing to someone who is not familiar with these methods (learned convolutional features, fully-connected hidden layers, ReLU rectified, convolutional weight matrices, training, testing, validation data). One should be able to understand the concept with Fig. 2 without having to read the methods section.

Response to reviews of: Deep learning to infer eddy heat fluxes from sea surface height patterns of mesoscale turbulence

NCOMMS-19-539539A

Authors: Tom George, Georgy Manucharyan & Andrew F. Thompson

Summary statement to the editor and reviewers

First, we thank the editor for providing sufficient time to reply to the most recent round of reviews – it has been a busy period for all of the authors. We have now had an opportunity to revisit comments from both rounds of reviews and we believe that we can adequately respond to the remaining concerns. It may also be helpful to briefly summarize the status of the paper. The first draft received three reviews, and our replies and the revised manuscript were considered largely satisfactory to both reviewers 1 and 2. Reviewer 3 had a more significant concern about our choice to focus this study on learning the system’s heat flux, as opposed to the divergent component of the heat flux or the heat flux divergence itself. Reviewer 3 was unable to review our revised manuscript, which was sent to a fourth reviewer who largely upheld the comments of Reviewer 3. After further reading of the literature and completing additional diagnostics of the model, we believe we can respond to both Reviewers 3 and 4 (this also addresses a lingering concern of Reviewer 2) in two different ways:

1. First, a direct suggestion from the reviewers was to show that our deep learning approach could predict the heat flux divergence. This required substantial additional work, and we now show in the manuscript that the CNN technique can indeed predict the heat flux divergence with roughly the same prediction skill as the heat flux. However, while this may satisfy the reviewers’ concerns, we feel strongly that it is inappropriate to pivot the manuscript’s focus to the heat flux divergence because the nature of the model (homogeneous QG turbulence) means that the heat flux divergence that we simulate is somewhat trivial. Specifically, the model itself is designed to have zero domain-averaged heat flux divergence, and only a weak heat flux divergence arises when considering subdomains. This actually makes it rather remarkable that the CNN can learn the heat flux divergence, but it also implies that the CNN success would not necessarily translate to more realistic flow regimes. We provide additional details below in the replies to the reviewers.
2. We feel that a more compelling argument for responding to Reviewers 3 and 4 is that (again by the nature of the model) the total heat flux in the baroclinic QG simulations is dominated by the divergent component. We agree that in real-ocean situations, distinguishing between rotational and divergent components will be critical for applying machine learning techniques to diagnosing eddy heat fluxes. However, this is not a limitation of our study because the rotational component is weak (at sufficiently large scales). To support this argument, we appeal to the eddy potential energy (EPE) budget, following Marshall and Shutts (1981), and show that advection of EPE is a small term in the budget. As shown in Marshall and Shutts (1981), it is this advection of EPE that gives rise to the rotational fluxes. We certainly agree that this will be a topic that future approaches to using machine learning for diagnosing eddy heat fluxes will need to address.

Finally, we note that Reviewer 3 originally offered three paths to addressing his concerns, quoted below in red. We feel we have now addressed all three:

1. **Redoing the paper with heat flux divergence as the target for learning.** We have completed this activity, please see figure 4 in the revised manuscript. We have found that our CNNs can indeed learn the heat flux divergence, however, this divergence may not be a good representation of real-ocean flows.
2. **Eliminating the sub-region averaging and redoing the experiment for the whole domain (global average eliminates rotational flux.)** We have shown that our subdomains are large enough that they do not have a significant rotational flux component.

3. Making a convincing and novel argument why the spatially averaged raw meridional heat flux is in fact a suitable target for machine-learning inference. For example, demonstrating how this quantity appears in a heat budget or other prognostic conservation equation. We derive the eddy potential energy budget below and show that the eddy heat flux is a key component of this budget.

We feel that our revised manuscript addresses the remaining major criticisms of the paper on both a direct, if somewhat superficial, level (successful deep learning of the heat flux divergence) and on a more fundamental level (a defense of the heat flux as a key component of the energy budget in the 2-layer QG model). We provide detailed responses to the reviewers' comments in the following replies. We are grateful to all four reviewers – all of their comments have been insightful and respectfully presented, and have pushed us to deepen our understanding of this study. We hope that the reviewers will find our following responses adequate, but we are happy to provide further clarification if necessary. Reviewer comments are in blue; our replies are in black.

Response to reviewer 1:

Thank you for addressing my concerns in your revised manuscript. I believe it is now suitable for publication. Thank you, once again, for making your dataset and code public - this is crucial for a community-driven investigation into scientific machine learning for the geophysical community.

Thank you for your time in reading the revised manuscript as well as for your initial comments.

Response to reviewer 2:

The authors adequately addressed all my specific comments. I recommend publication after a minor revision.

My remaining criticism of the paper is the focus on the total eddy heat flux. The main argument is that once the full flux is known, one can perform the Helmholtz decomposition and extract the divergent part. There are two problems with this logic. Firstly, extraction of the divergent flux component is not unique, because it strongly depends on the lateral boundary conditions for the divergent component, which are not uniquely defined. Another problem is that the total eddy heat flux tends to be dominated by the rotational component, and even a relatively small error in the total flux can lead to large errors in the divergent part.

Dear Igor – thank you for your time in reviewing our revised version of the manuscript and for these additional comments. You raise an important concern regarding the fact that, for rotationally-dominated flows such as ours, a small error in the total heat flux may result in a large error in the extraction of the divergent component (and therefore the domain-averaged flux divergence). As presented in our summary statement above, we can respond to this concern in two ways. First, and perhaps most importantly, while it is generally true in the ocean that the heat flux is often dominated by a rotational component, we do not believe that this is the case for our homogeneous two-layer turbulence simulations. We present a defense of this statement in our reply to Reviewer 4 (we have asked the editor to make all our replies available to each reviewer). We have additionally shown in the paper that the CNN does indeed appear to be able to learn the heat flux divergence reasonably well. We feel that this will allay some reader concerns and therefore, we have added a new subsection and figure to the manuscript that discusses analysis of the heat flux divergence. However, we note that this divergence is unlikely to be closely related to processes that generate heat flux divergence in the ocean due to the nature of the QG model. Therefore, we believe it is more appropriate to maintain the focus of the study on the eddy heat flux.

In the manuscript, we have also provided additional clarifying statements about why the eddy heat flux (and not its divergence) affects the evolution of the eddy kinetic energy within a domain. This result has been confirmed theoretically and experimentally [1,2]. We have now checked that this result holds for our synthetic data and we have added a curve to Manuscript Figure 1e (see Response Figure 2 at the end of this document) showing the strong, slightly delayed, correlation between the heat flux and the kinetic energy. We also motivate the study by arguing that some of the most ambitious efforts of the observational oceanographic community over the past decades have involved the design and implementation of observing systems that can directly measure volume and heat transport, via fluxes, across a single (zonal) transect. Examples include the RAPID [3] and OSNAP [4] programs. These observing arrays have provided critical new understanding about the temporal (and increasingly spatial) variability of the Atlantic Meridional Overturning Circulation (AMOC), but at a cost of many tens of millions of dollars. If comparable estimates could be achieved using remote sensing products, and across a larger number of latitudes, it would represent a step change in how we observe ocean circulation. It is clear therefore that our understanding of oceanic energy transfers due to baroclinic instabilities would greatly benefit from deep learning methods which can generate accurate time series of the heat flux. The above arguments and references to the RAPID/OSNAP programs have been added to the manuscript.

We also acknowledge that we remain far from the end goal of using deep learning as training will be difficult from observations alone! Nevertheless, we believe that our study indicates that there is value in continuing to pursue this path.

On the other hand, the paper describes a promising method of using SSH to diagnose heat flux – a nonlinear quantity that depends on the 3D flow and temperature structure. One can argue that the degree to which the quantity itself (full heat flux) is critical for describing the temperature evolution is of secondary importance here.

While we hope that our arguments are convincing that the heat flux is not necessarily of secondary importance, we do agree that that a major result of the study is that CNNs can provide insight into the link between SSH variations and eddy flux properties. We hope to show that CNNs (or, more generally, deep learning) are powerful methods for estimating non-linear depth-dependent quantities in the ocean. We believe that basic deep learning should be added to the tool-box of physical oceanographers and that this paper shows one such useful implementation. We hope that this paper will provide motivation for further studies in diagnosing eddy fluxes – and flux divergences – using deep learning.

I recommend that the authors move the discussion of the importance of the divergent heat flux (“It is well known that . . .”) from the end of the Methods section to either the Introduction or Discussion, and expand it, clearly stating that although the results may not lead to accurate estimates of instantaneous heat flux divergence, the main point of the paper is to demonstrate that the method works for using SSH to estimate a nonlinear quantity that depends on 3D velocity and temperature.

We have added to the Discussion, as you suggest, a sentence stating that one of the main points of the paper is the technique itself, not the precise quantity we train on. In the new Results subsection, which discusses heat flux divergence prediction using CNNs, we have now mentioned if the CNN-derived estimates of instantaneous heat fluxes have limitations in estimating the instantaneous heat flux divergence (for example, if the rotational component is large), this can potentially be bypassed by learning the heat flux divergence instead. In this section and the Introduction we further clarify the importance of both the heat flux and the heat flux divergence.

A direct comparison between the CNN-estimated and actual flux divergence would, of course, be even more convincing, but I expect the corresponding biases to be large.

Since receiving this round of reviews we have done almost exactly this. Instead of comparing the divergence of the flux predicted by the CNN (which as you rightly suggest can potentially lead to large errors) we trained a CNN to learn the heat flux divergence outright, from the SSH input. Response Figure 1 (see end of this response document) (Manuscript Figure 4) shows the results of attempting this. A ‘flux boundary’ is defined within the 1000×1000 km subdomain and the net flux flowing across this boundary is calculated (subpanel a). The CNN is then trained to estimate the divergence in the subdomain given the corresponding SSH snapshot; note that this subdomain is still large enough to contain multiple coherent eddies. When the flux divergence boundary is inset from the full subdomain by ~ 200 km – so that eddies crossing the boundary, which would cause large divergence fluctuations, are fully visible – the CNN can learn to estimate the heat flux divergence as accurately as it can learn to estimate the heat flux (e.g. skill = 0.35 vs original skill = 0.36) reported in the initial manuscript (subpanels b and c). We recommend comparing Response Figure 1d (now Manuscript Figure 4d) to Manuscript Figure 3d in order to see that the reconstruction of the flux divergence is as good as the reconstruction of the flux.

This figure, and a dedicated subsection covering how the CNN can be used to estimate the flux divergence, has been added to the Results. However, we keep this section relatively brief because of the limited “reality” of the heat flux divergence field.

Response to Reviewer 4:

First, we appreciate the time of Reviewer 4 in considering the manuscript in the context of Reviewer 3’s earlier comments. These additional comments have forced us to readdress the goals of our study and the information contained in the 2-layer QG simulations. We are able to respond to the major criticism regarding the study’s focus on the eddy heat flux, rather than the heat flux divergence, in two ways as summarized above.

I agree that it is the divergence of the eddy heat flux that should have been considered in the first place. I don’t think we are mainly interested in how well the variance of the eddy heat flux can be predicted (e.g. the ”wiggles” of alternating large and small heat fluxes), but we want to know what is the net effect, i.e. how much heat is fluxed in and out of a region.

We agree that there is value in both learning the eddy heat flux and the heat flux divergence; both diagnostics provide different information of the flow. We also agree that the heat flux divergence is most useful for constraining the heat budget, but the heat flux itself is an important term in the energy budget. Neither of these budgets is “more important” than another.

Nevertheless, a direct response to the comment above is to attempt to build a CNN that can also learn the heat flux divergence. As our first step in responding to these reviews we did exactly this. The results of this analysis are now presented in Response Figure 1 (Manuscript Figure 4) at the end of this document. First, within our doubly periodic domain, we considered 1000×1000 km subdomains and within each subdomain defined a “flux boundary” across which the net diverging flux is calculated (subpanel a). The CNN is then trained to estimate this subdomain divergence given the full corresponding SSH snapshot. Note that this subdomain is still large enough to contain multiple coherent eddies. When the flux boundary is inset from the full subdomain by 200 km the CNN can learn to estimate the heat flux divergence as accurately as it can learn to estimate the heat flux (e.g. skill = 0.35 vs original skill = 0.36) reported in the initial manuscript (subpanels b and c). The reason for defining a flux boundary that is inset within the subdomain is that eddies crossing the boundary, and extending beyond the flux boundary, provide the dominant contribution to the flux divergence fluctuations. Comparing Response Figure 1c (now Manuscript Figure 4c) to Manuscript Figure 3d, we find that the reconstruction of the flux divergence is as good as the reconstruction of the flux. Response Figure 1, and a short subsection covering how the CNN can be used to estimate the flux divergence, has been added to the Results to address the reviewers’ comments.

However, after further consideration of this work, we were uncomfortable with re-focusing our study on the heat flux divergence using this QG model. Specifically, the eddy heat flux, which in the Phillips model is proportional to the upper layer PV flux and energy production, has zero divergence when averaged over the domain. Thus, the heat flux divergence that is diagnosed in the subdomains described above is likely to arise from physics that are quite different from the processes in the ocean that lead to heat flux divergence.

Thus, we returned to the issue of rotational and divergent components of the heat flux in the Phillips model to convince ourselves (and hopefully the reviewers) that the eddy heat flux diagnosed from the model is a dynamically-relevant quantity. We acknowledge that for general ocean flows, developing deep learning techniques that can distinguish between these components of the heat flux will be essential – the concerns of both Reviewers 3 and 4 are valid. However, due to the nature of our model, we feel it is incorrect to suggest that the heat fluxes learned by our CNNs are meaningless because they are almost identically equal to the divergent component of the heat flux. To see this, we refer Reviewer 4 to Marshall and Shutts (1981) and first reproduce their equation for the steady state eddy potential energy (EPE) equation:

$$\bar{v} \cdot \nabla \frac{\overline{T'}}{2} + \overline{v'T'} \cdot \nabla \bar{T} + \overline{w'T'} \frac{\partial \bar{T}}{\partial z} = 0. \quad (1)$$

This equation relates the eddy flux of heat across the mean temperature gradient (second term) to the

conversion rate of EPE to eddy kinetic energy (third term) and the advection of EPE by the mean flow (first term). The term that we diagnose from our model is equivalent to the eddy heat flux (second term). Marshall and Shutts show that the eddy heat flux $\overline{v'T'}$ can be decomposed into divergent and rotational components. In their equation (3), they further show that the divergent (subscript D) and rotational (subscript R) components are balanced by different terms in the EPE budget:

$$(\overline{v'T'})_D \cdot \nabla \overline{T} + \overline{w'T'} \frac{\partial \overline{T}}{\partial z} = 0, \quad (2)$$

$$(\overline{v'T'})_R \cdot \nabla \overline{T} + \overline{v} \cdot \nabla \frac{\overline{T'}}{2} = 0. \quad (3)$$

A critical statement of (3) is that the rotational component of the eddy heat flux arises from a need to balance the advection across a gradient in the EPE distribution. This advection term is often significant in the ocean in situations where strong mean flows, like the Gulf Stream, become baroclinically unstable and then advect mesoscale eddies downstream. However, this is different to the turbulence generated in our 2-layer *homogeneous* baroclinic QG model, which by design has no gradients in EPE at the domain scale. Granted, these gradients may arise within the subdomains, but as long as these domains are significantly larger than the eddy scale, they should also be largely homogeneous. Thus, if we can show that the rotational component is small, the eddy heat flux we diagnose and predict with our CNNs represents the divergent component of the heat flux.

This next step requires us to write down the heat equation for the 2-layer QG model and then form the EPE budget. We start from an expression for the evolution of the layer thicknesses:

$$\frac{Dh_i}{Dt} = \frac{\partial h_i}{\partial t} + J(\psi_i, h_i) = (-1)^i w, \quad (4)$$

where $i = 1, 2$ indicates the layer (1=upper and 2=lower) and w is the vertical velocity associated with the interface displacement. Using the relationship $h'_1 + h'_2 = 0$ (where primes indicate deviations away from the mean and noting the total water column depth is fixed), we can take the difference of the two equations in (4) to arrive at:

$$\frac{\partial}{\partial t} (2h'_1) + J(\psi_1 + \psi_2, h'_1) = -2w. \quad (5)$$

The last step is to relate the interface displacement to the streamfunctions and therefore the temperature. In our simulations, we use unequal layers depths, which is more realistic for the ocean, but makes the derivation of what follows a bit clumsy. For now, we proceed assuming equal layer depths $H_1 = H_2$ and $\delta = H_1/H_2 = 1$, which allows an easier comparison to (1); below we add the additional pre-factors that arise from unequal layer depths. Via geostrophy we have

$$h_1 = \frac{f}{g'} (\psi_1 - \psi_2) = \frac{2f}{g'} \tau, \quad (6)$$

where we used the fact that the baroclinic mode, $\tau = (\psi_1 - \psi_2)/2$ (for equal layer depths), is proportional to the temperature anomaly in the 2-layer QG model (see a detailed discussion of this point in Thompson and Young 2006). Plugging (6) into (5), we arrive at:

$$\frac{\partial}{\partial t} \tau + J(\psi, \tau) = -\frac{g'w}{2f}. \quad (7)$$

For equal layer depths, $\psi = (\psi_1 + \psi_2)/2$ is the barotropic streamfunction and we assume the background mean barotropic flow is U . To form the EPE evolution equation, we multiply (7) by τ :

$$\frac{\partial}{\partial t} \left(\frac{\tau^2}{2} \right) + J \left(\psi', \frac{\tau^2}{2} \right) + U \frac{\partial}{\partial x} \left(\frac{\tau^2}{2} \right) - (\psi'_x \tau) U + (w\tau) \frac{g'}{2f} = 0. \quad (8)$$

The similarity between (1) and (8) is clear. Assuming steady state, the advection of temperature variance by both the eddy and mean barotropic velocities (terms 2 and 3) balance the horizontal heat flux (term 4, where $-U$ is related to the horizontal temperature gradient through thermal wind balance) and the conversion of kinetic energy to potential energy (term 5). Following equations (2) and (3), it remains to show that the advection of tracer variance is small compared to the eddy heat flux.

In the simulations discussed in the manuscript, we have a shallow upper layer and a thicker lower layer to represent a pycnocline at roughly 1000 m depth, such that $\delta = 0.2$. We make the following adjustments following Flierl (1978) and Arbic and Flierl (2004):

$$h_1 = \frac{2}{1+\delta} \frac{f}{g'} (\psi_1 - \psi_2) = \frac{2}{\sqrt{\delta}} \frac{f}{g'} \tau, \quad \tau = \frac{\sqrt{\delta}}{1+\delta} (\psi_1 - \psi_2), \quad \psi = \frac{\delta\psi_1 + \psi_2}{1+\delta}. \quad (9)$$

With these additional factors of δ , (8) becomes

$$\frac{\partial}{\partial t} \left(\frac{\tau^2}{2} \right) + J \left(\psi'_1, \frac{\tau^2}{2} \right) + J \left(\psi'_2, \frac{\tau^2}{2} \right) + U \frac{\partial}{\partial x} \left(\frac{\tau^2}{2} \right) - \frac{\sqrt{\delta}}{1+\delta} U (\psi'_1 + \psi'_2)_x \tau + (w\tau) \frac{\sqrt{\delta}g'}{2f} = 0. \quad (10)$$

In the plots below, we assess the magnitude of the terms in (10) with the proper layer-depth ratio. Note that the QG model does not explicitly calculate the vertical velocity w , but it can be diagnosed from (5). Indeed by substituting (5) into (10) in place of w in the last term, the preceding five terms in (10) are recovered with the opposite sign. This shows that the eddy heat flux in the QG model can indeed be partitioned as in (2) and (3).

In Figure 3 we plot the magnitude of the terms that contribute to the advection of EPE (terms 2, 3 and 4) and the eddy heat flux (term 5) in eq. (10), evaluated in a 1000×1000 km subdomain for one of our equilibrated QG simulations. The eddy heat flux is at least an order of magnitude larger than both of these terms. We have also tested the sensitivity of this difference to the size of the subdomain, as shown in Figure 4a. For subdomains that are 1000 km a side or larger, the ratio of the variance in EPE advection to eddy heat flux is small. Together, these plots suggest that only a small component of the eddy heat flux can be explained by the rotational component. We also show that our subdomain size, $1000 \text{ km} \times 1000 \text{ km}$, is sufficiently large to minimize error in the estimate of the divergent heat flux (Figure 4b).

To summarize, the derivation above confirms that the eddy heat flux (and not its divergence) directly affects the evolution of the eddy energy budget within a domain and hence warrants being the focus of this paper. This result is consistent with previous theoretical and experimental studies [1,2]. We have also added a curve to Manuscript Figure 1e (see Response Figure 2) showing the strong, slightly delayed, correlation between the heat flux and the kinetic energy. We have also added an additional motivation statement highlighting that the largest, most ambitious, and arguably most important efforts of the observational oceanographic community over the past decades have involved the design and implementation of observing systems that can directly measure volume and heat transport, via fluxes, across a single (zonal) transect. Examples include the RAPID [3] and OSNAP [4] programs. These observing arrays have provided critical new understanding about the temporal (and increasingly spatial) variability of the Atlantic Meridional Overturning Circulation (AMOC), but at a cost of many tens of millions of dollars. If comparable estimates could be achieved using remote sensing products, and across a larger number of latitudes, it would represent a step change in how we observe ocean circulation. It is clear therefore that our understanding of oceanic energy transfers due to baroclinic instabilities would greatly benefit from deep learning methods which can generate accurate time series of the heat flux. The above arguments and references to the RAPID/OSNAP programs have been added to, or clarified within, the manuscript.

You say that once you have the heat flux, then the divergence can always be computed, but if you have the divergence, then the heat flux cannot be uniquely recovered, which is of course true. The question is are your

algorithms “conservative” in the sense that they provide the eddy heat flux that will give you the right divergence? As Fig. 3 shows, while it’s true that the “wiggles” in time are quite nicely reproduced by the pattern recognizing algorithms, the magnitude of the heat flux, i.e. the net heat flux over the subdomains, is not. Unless I misunderstood something, in the supplementary material you provide calculations saying that the divergence of the averaged eddy heat fluxes equals the averaged eddy heat flux divergence. But that is trivial.

These are all valid concerns and, as also suggested by reviewer 2, in scenarios where the rotational component is much larger than the divergent component of the heat flux, small errors in the predicted heat flux could give rise to large errors in the heat flux divergence (if calculated directly from the predicted heat flux). And yes, we agree that one can not uniquely recover the heat flux from the divergence alone. We believe that our response to the previous comments addresses most of the concerns here: in our model, the heat flux is dominated by the divergent component. The fact that we find that CNNs can learn the heat flux divergence in this model is promising that it could be used to learn the heat flux divergence in more realistic (and potentially rotationally-dominated situations), but this requires further work and is beyond the scope of this study.

You should compare the divergence of the predicted eddy heat flux with the divergence of the true eddy heat flux. What would the time series of Fig. 3d look like if you compared the predicted and true divergence? What does the bias in the magnitude of the eddy heat flux do to the divergence of the eddy heat flux? A related question is how does this bias depend on the size of the subdomain? The eddy heat flux locally is very noisy, very large values are next to very small values, the subdomain-averaged eddy heat flux for a smaller subdomain is very different from the subdomain average over a larger domain. All of these questions can be avoided by predicting the divergence of the eddy heat flux in one subdomain.

We followed the Reviewer’s recommendation here and found that the CNNs could indeed be successfully trained to learn the heat flux divergence (Figure 1, or manuscript figure 4). We have reservations making this the major focus of the study for the reasons discussed in detail above. We hope, however, that these results provide evidence that CNNs are indeed valuable tools to pursue and that the results of our study will be of broad interest to the machine learning and geophysical communities. This result and a new figure is added as a short additional subsection in the Results.

I still think that the CNN method is not explained well-enough for an audience that does not know much or anything about “deep learning” algorithms. The authors should make another effort to explain the principle, before delving into Fig. 2. Also, Fig. 2 is still full of terms that mean nothing to someone who is not familiar with these methods (learned convolutional features, fully-connected hidden layers, ReLU rectified, convolutional weight matrices, training, testing, validation data). One should be able to understand the concept with Fig. 2 without having to read the methods section.

We acknowledge that these terms may not be as familiar to an oceanography audience but they are basic concepts in the field of deep learning. Nonetheless, a balanced compromise must be found and we have revised the manuscript to improve our description of the architecture and some deep learning terminology to the point where a person knowledgeable of only the very basics of deep learning and CNNs should be able to understand the architecture and why we have used it. To ensure the reproducibility of our results, we have provided our Python scripts and datasets.

Furthermore, to provide a reader with limited machine learning experience an intuition behind our choice of CNNs (even if they do not strive to understand the architecture specifics) the following paragraph has been added to the Introduction, before discussion of Figure 2. It provides a layman explanation of CNNs:

Here we use Convolutional Neural Networks (CNNs), a class of ANNs for which the input is a 2D image. Historically CNNs have been successful at solving image/pattern recognition problems for which no analytical solution exists[5]. Like all “deep” neural networks they are built from simple layers stacked atop one another

between non-linear functions. For the CNN each layer filters the output of the layer before by convolving a small filter matrix across it. Both the depth and non-linearity of the resulting model are key to explaining its strength. CNN filters are not specified a priori but instead are optimized from the data until they minimize an objective function using a gradient descent algorithm. This provides a path towards solving analytically intractable problems, such as distinguishing images of cats and dogs or predicting oceanic heat fluxes from sea surface height measurements.

The caption for figure 2 has also been rewritten for clarity.

There remains an outstanding issue from the first round of reviews, regarding an analysis carried out by Reviewer 3. Here, Reviewer 3 (Ryan Abernathey, signed review) completed a Helmholtz decomposition of the heat flux from our model output and concluded that the divergent component of the flux was small. We respond to this point here, although it was not explicitly raised again by Reviewer 4.

A key point is that the dynamically-relevant component of the heat flux is spatially-uniform at the scale of our $1000 \text{ m} \times 1000 \text{ m}$ subdomains. The Helmholtz decomposition involves taking the divergence of the flux as an initial step, and thus removes the spatially-uniform component.

References

- Arbic, B. K., and G. R. Flierl, 2004. Baroclinically unstable geostrophic turbulence in the limits of strong and weak bottom Ekman friction: Application to midocean eddies. *J. Phys. Oceanogr.*, **34**, 2257–2273.
- Flierl, G. R., 1978. Models of vertical structure and the calibration of two-layer models. *Dyn. Atmos. Oc.*, **2**, 341–381.
- Marshall, J., and G. Shutts, 1981. A note on rotation and divergent eddy fluxes. *J. Phys. Oceanogr.*, **11**, 1677–1680.

Figure 1: CNNs are capable of learning the eddy heat flux divergence (HFD) into/out of a domain. (a) The flux divergence is defined as the total ‘non-trivial’ eddy heat flux flowing across the flux divergence boundary. (b) Training curve showing the training, validation and test error of the CNN as it learns the heat flux divergence for boundary inset $x = 200$ km. (c) CNN performance for a range of flux divergence boundary insets. The CNN can optimally learn to predict HFD when the boundary is at least 200 km smaller than the domain it receives as input. It achieves a skill of 0.35, equal to the performance of the CNN trained to learn the flux itself (paper figure 3). (d) Time series showing the fluctuations of the true and predicted heat flux divergence (boundary inset $x = 200$ km). This figure appears in the updated manuscript

Figure 2: Manuscript figure 1e has been updated to include the blue curve. Lag correlation curves show that the heat flux (not the heat flux divergence) correlates strongly with the kinetic energy (blue) with the strongest correlation occurring after 10 days – the heat flux itself is therefore a useful quantity to predict as it contains information about the total kinetic energy within the domain. showing the correlation between the subdomain average heat flux (not heat flux divergence) and the subdomain average kinetic energy. The negative sign arises since (for our simulation) the baroclinic instability generates a negative meridional heat flux (see manuscript figure 1d) - when this increases the magnitude of the fluxes fall and so too does the kinetic energy giving the negative correlation.

Figure 3: Time series of terms contributing to the eddy potential energy budget. The values are computed within a 1000×1000 km subdomain of a larger doubly-periodic simulation. We show that the eddy heat flux (and its variance, Figure 4) is an order of magnitude larger than the advection of eddy potential energy by both eddies (blue/red curves) and the mean flow (yellow curve).

Figure 4: Analysis of eddy potential energy (EPE) advection. (a) Ratio of the variance associated with the advection of EPE to the eddy heat flux as a function of subdomain size. This suggests that eddy heat flux is dominated by the divergent component following Marshall and Shutts (1981). (b) Estimate of the error in the divergent component of the eddy heat flux due to advection of EPE; this error is small for a sufficiently large subdomain. In our study we use a subdomain of 1000 km, indicated by the dashed lines in each panel.

Reviewer #2 (Remarks to the Author):

The authors fully addressed my previous concerns. In my opinion, the manuscript is now ready for publication.

Igor Kamenkovich

Reviewer #4 (Remarks to the Author):

The authors have addressed the issue of discussing the relevance of constraining the heat flux vs the divergence of the heat flux. I appreciate the inclusion of an additional section where the divergence of the heat flux is constrained and appreciate the significant additional work that went into it. So it's great to see that this works, too and I still think it is this quantity that should be constrained from observations since there will be significant errors if the divergence is computed from the estimated heat fluxes themselves (the comparison of the divergence of the predicted heat flux with the actual divergence you did not show). In the real ocean, the eddy heat flux is always dominated by the rotational parts, at least on scales governed by QG dynamics, and even in observational campaigns as RAPID or OSNAP it is a 'net' basin-wide heat transport that is of interest.

I'm ok with publication now, just would like in line 105 a caveat added:

In the absence of a statistically-significant advection of EPE by the mean flow "and by the eddies", the rotational flux is negligible, "at least when the subdomain over which the average is taken is sufficiently large". Also, please add the caveat that the Helmholtz decomposition is not unique and there is never really "the" rotational and "the" divergent part.